# Nonlinear stimulus representations in neural circuits with approximate excitatory-inhibitory balance

Cody Baker[1], Vicky Zhu[1], Robert Rosenbaum[1,2]*

**1** Department of Applied and Computational Mathematics and Statistics, University of Notre Dame, Notre Dame, IN, USA, **2** Interdisciplinary Center for Network Science and Applications, University of Notre Dame, Notre Dame, IN, USA

* Robert.Rosenbaum@nd.edu

## Abstract

Balanced excitation and inhibition is widely observed in cortex. How does this balance shape neural computations and stimulus representations? This question is often studied using computational models of neuronal networks in a dynamically balanced state. But balanced network models predict a linear relationship between stimuli and population responses. So how do cortical circuits implement nonlinear representations and computations? We show that every balanced network architecture admits stimuli that break the balanced state and these breaks in balance push the network into a "semi-balanced state" characterized by excess inhibition to some neurons, but an absence of excess excitation. The semi-balanced state produces nonlinear stimulus representations and nonlinear computations, is unavoidable in networks driven by multiple stimuli, is consistent with cortical recordings, and has a direct mathematical relationship to artificial neural networks.

## Author summary

Several studies show that neurons in the cerebral cortex receive an approximate balance between excitatory (positive) and inhibitory (negative) synaptic input. What are the implications of this balance on neural representations? Earlier studies develop the theory of a "balanced state" that arises naturally in large scale computational models of neural circuits. This balanced state encourages simple, linear relationships between stimuli and neural responses. However, we know that the cortex must implement nonlinear representations. We show that the classical balanced state is fragile and easily broken in a way that produces a new state, which we call the "semi-balanced state." In this semi-balanced state, input to some neurons is imbalanced by excessive inhibition—which transiently silences these neurons—but no neurons receive excess excitation and balance is maintained the sub-network of non-silenced neurons. We show that stimulus representations in the semi-balanced state are nonlinear, improve the network's computational power, and have a direct relationship to artificial neural networks widely used in machine learning.

**Data Availability Statement:** All data and code to produce all figures can be found at https://github.com/RobertRosenbaum/SemiBalanceNets/.

**Funding:** RR was supported by National Science Foundation grants DMS-1654268 and Neuronex

DBI-1707400 as well as an award by the Huisking Foundation, Inc. The funders had no role in study design, data collection and analysis, decision to publish, or preparation of the manuscript.

**Competing interests:** The authors have declared that no competing interests exist.

## Introduction

An approximate balance between excitatory and inhibitory synaptic currents is widely reported in cortical recordings [1–6]. The implications of this balance are often studied using large networks of model neurons in a dynamically stable balanced state. Despite the complexity of spike timing dynamics in these models, the statistics of population responses to stimuli are described by a relatively simple and widely studied mean-field theory [7–18].

However, the classical theory of balanced networks has at least two shortcomings. First, it predicts a linear relationship between stimuli and neural population responses, in contrast to the nonlinear computations that must be performed by cortical circuits.

Secondly, parameters in balanced network models must be chosen so that the firing rates predicted by balanced network theory are non-negative. In the widely studied case of one excitatory and one inhibitory population, parameters for network connectivity and external input must satisfy only two inequalities to achieve positive predicted rates [8, 11]. However, strictly positive predicted rates can be more difficult to achieve in networks with several populations such as multiple neuron subtypes, neural assemblies, or tuning preferences [13, 19]. This difficulty occurs because the proportion of parameter space for which predicted rates are non-negative becomes exponentially small with an increasing number of populations. Moreover, a given network architecture might produce a balanced state for some stimuli, but not others. Indeed, we show that for any network architecture satisfying Dale's law, there are infinitely many excitatory stimuli for which balanced network theory predicts negative rates, implying that any network structure admits stimuli that break the classical balanced state.

We address these problems with balanced network theory by developing a theory of semi-balanced networks that quantifies network responses when the classical balanced network state is broken. In the semi-balanced state, balance is only enforced in one direction: neurons can receive excess inhibition, but not excess excitation. Neurons receiving excess inhibition are silenced and the remaining neurons form a balanced sub-network. We show that semi-balanced networks implement nonlinear stimulus representations and computations. Specifically, we establish a mathematical relationship between semi-balanced networks and artificial neural networks used for machine learning [20], as well as threshold-linear networks studied for their rich dynamics [21–24]. We show that semi-balance, but not balance, is naturally realized at a neuron-by-neuron level in networks with homeostatic inhibitory plasticity [25, 26] and time-varying stimuli. In this setting, semi-balanced networks implement richly nonlinear stimulus representations. We demonstrate the computational capacity of these representations on the hand-written digit classification benchmark, MNIST.

In summary, in contrast to the classical balanced state, the semi-balanced state is realized naturally in networks with time-varying stimuli, produces nonlinear stimulus representations, and has a direct correspondence to artificial neural networks used in machine learning. The theory of semi-balanced networks therefore has extensive implications for understanding stimulus representations and computations in cortical circuits.

## Results

### Balanced networks implement linear stimulus representations and computations

To review balanced network theory and its limitations, we consider a recurrent network of $N = 3 \times 10^4$ randomly connected adaptive exponential integrate-and-fire (adaptive EIF) neuron models (Fig 1A). The network is composed of two excitatory populations and one inhibitory population (80% excitatory and 20% inhibitory neurons altogether) and receives

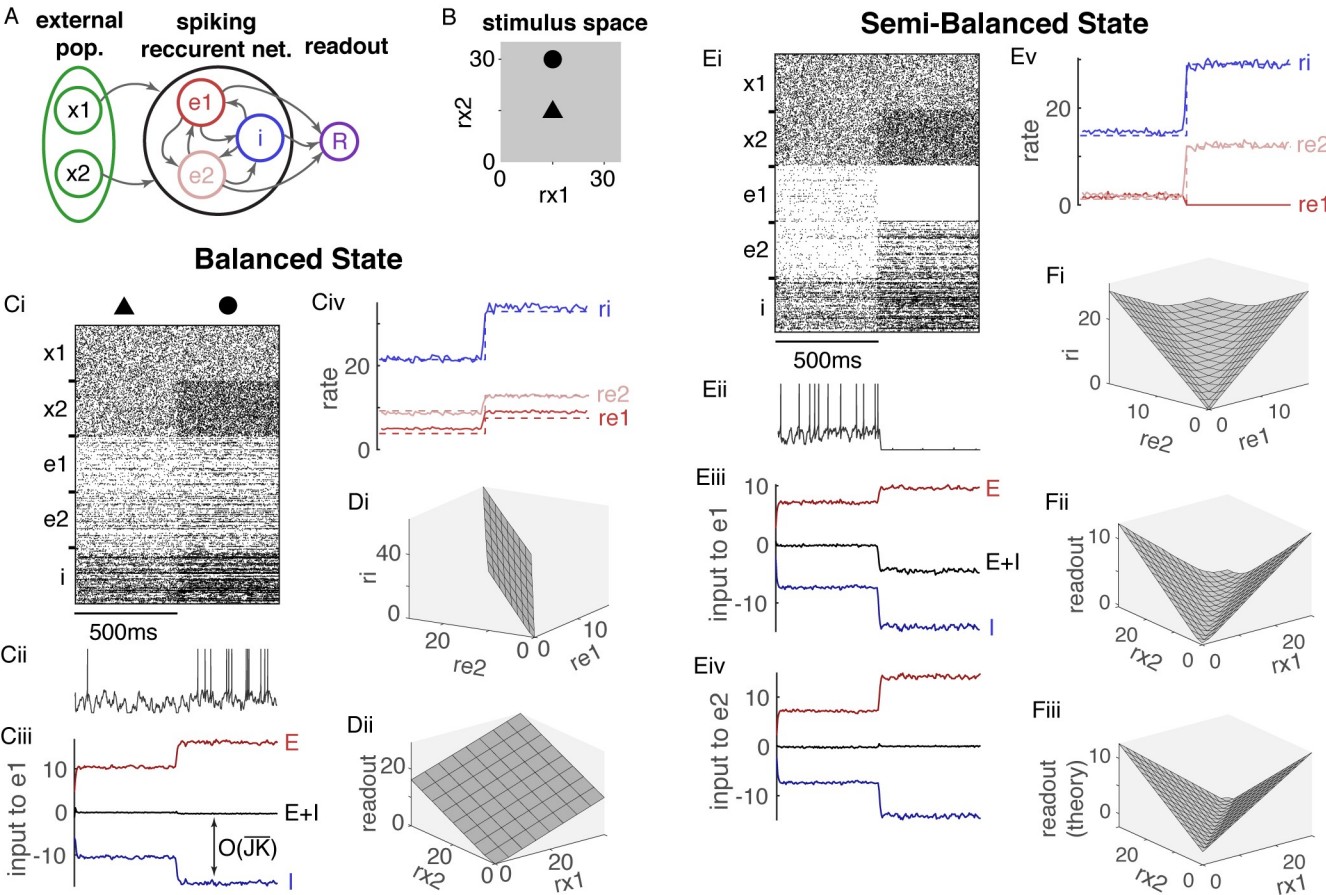

**Fig 1. Stimulus representations are linear in the balanced state and nonlinear in the semi-balanced state. A)** Network diagram. A recurrent spiking network of $N = 3 \times 10^4$ model neurons is composed of two excitatory populations ($e1$ and $e2$) and one inhibitory population ($i$) that receive input from two external spike train populations ($x1$ and $x2$). Recurrent network output is represented by a linear readout of firing rates ($R$). **B)** The two-dimensional space of external population firing rates represents a stimulus space. Filled triangle and circle show the two stimulus values used in C and E. **Ci)** Raster plots of 200 randomly selected spike trains from each population for two stimuli. **Cii)** Membrane potential of one neuron from population $e1$. **Ciii)** Mean input current to population $e1$ from all excitatory sources ($e1$, $e2$, $x1$, and $x2$; red), from the inhibitory population ($i$; blue), and from all sources (black) showing approximate excitatory-inhibitory balance across stimuli. Mean input to $i$ and $e2$ were similarly balanced. **Civ)** Firing rates of each population from simulations (solid) and predicted by Eq (3) (dashed). **Di)** The neural manifold traced out by firing rates in each population in the recurrent network as external firing rates are varied across a square in stimulus space ($0 \le r_{x1}, r_{x2} \le 30$). **Dii)** The readout as a function of $r_{x1}$ and $r_{x2}$ from the same simulation as Di. **Ei–v)** Same as Ai–iv, but dashed lines in Dv are from Eq (4) and input to $e2$ was additionally shown. D **Fi–iii)** Same as Di–ii except the theoretical readout predicted by Eq (4) was additionally included. All firing rates are in Hz. All currents are normalized by the neurons' rheobase.

feedforward synaptic input from two external populations of Poisson processes, modeling external synaptic input. The firing rates, $\boldsymbol{r}_x = [r_{x1} \ r_{x2}]^T$, of the external populations form a two-dimensional stimulus space (Fig 1B; $\boldsymbol{v}^T$ denotes the transpose of $\boldsymbol{v}$).

Simulations of this model showed asynchronous-irregular spiking activity and excitatory-inhibitory balance (Fig 1Ci–1Ciii). How does connectivity between the populations determine the mapping from stimulus, $\boldsymbol{r}_x$, to population-averaged firing rates, $\boldsymbol{r} = [r_{e1} \ r_{e2} \ r_i]^T$, in the recurrent network? Firing rate dynamics are often approximated using models of the form

$$\tau \dot{\boldsymbol{r}} = -\boldsymbol{r} + f(\overline{JK}[W\boldsymbol{r} + \boldsymbol{X}]) \tag{1}$$

where $\dot{\boldsymbol{r}}$ denotes the time derivative, $f$ is a non-decreasing f-I curve, and $W$ is the effective recurrent connectivity matrix. External input is quantified by $\boldsymbol{X} = W_x \boldsymbol{r}_x$. Components of $W$ and $W_x$ are given by $w_{ab} = J_{ab}K_{ab}/\overline{JK}$ where $K_{ab}$ is the mean number of connections

from population $b$ to $a$ and $J_{ab}$ is the average connection strength. The coefficient, $\overline{JK} = \text{mean}(|J_{ab}|K_{ab})$, quantifies coupling strength in the network. Since $\overline{JK}$ is multiplied in the equation for $\dot{r}$ and divided in the equation for $w_{ab}$, it does not affect dynamics, but serves as a notational tool to quantify the net strength of coupling in the network.

The key idea underlying balanced network theory is that $\overline{JK}$ is often large in cortical circuits because neurons receive thousands of synaptic inputs and each postsynaptic potential is moderate in magnitude. Total synaptic input,

$$I = \overline{JK}[W\boldsymbol{r} + \boldsymbol{X}], \tag{2}$$

can only remain $\mathcal{O}(1)$ if there is a cancellation between excitation and inhibition. In particular, to have $\boldsymbol{I} \sim \mathcal{O}(1)$, we must have $W\boldsymbol{r} + \boldsymbol{X} \sim \mathcal{O}(1/\overline{JK})$ so, in the limit of large $\overline{JK}$, firing rates satisfy [8, 15, 19, 27]

$$\boldsymbol{r} = -W^{-1}\boldsymbol{X}. \tag{3}$$

While Eq (1) is a heuristic approximation to spiking networks, Eq (3) can be derived for spiking networks and binary networks in the limit of large $\overline{JK}$ without appealing Eq (1) and even without specifying an f-I curve at all [8, 18, 28] Classical balanced network theory specifically considers the $K_{ab} \to \infty$ limit (with $N \to \infty$ where $N$ is the number of neurons in the recurrent network) while taking $J_{ab} \sim 1/\sqrt{K_{ab}}$ so that $\overline{JK} \to \infty$. Evidence for this scaling has been found in cortical cultures [6].

Even though it is derived as a limit, Eq (3) provides a simple approximation to firing rates in networks with finite $\overline{JK}$. Indeed, it accurately predicted firing rates in our spiking network simulations (Fig 1Civ, compare dashed to solid) for which $\overline{JK} = 5.9$ mV/Hz.

While the simplicity of Eq (3) is appealing, its linearity reveals a critical limitation of balanced networks as models of cortical circuits: Because $\boldsymbol{r}$ depends linearly on $\boldsymbol{X}$ and $\boldsymbol{r_x}$, balanced networks can only implement linear representations of stimuli and linear computations [8, 15, 27].

To demonstrate this linearity in our spiking network, we sampled a square lattice of $\boldsymbol{r_x}$ values and plotted the resulting neural manifold traced out in three dimensions by $\boldsymbol{r}$. The resulting manifold is approximately linear, i.e., a plane (Fig 1Di) because $\boldsymbol{r}$ depends linearly on $\boldsymbol{X}$, and therefore on $\boldsymbol{r_x}$, in Eq (3). More generally, the neural manifold will be an $n_x$-dimensional hyperplane in $n$-dimensional space where $n$ and $n_x$ are the number of populations in the recurrent and external populations respectively. In addition, any projection, $R = \boldsymbol{w} \cdot \boldsymbol{r}$, is a linear function of $\boldsymbol{r_x}$ and therefore also planar (Fig 1Dii).

This raises the question of how cortical circuits, which exhibit excitatory-inhibitory balance, can implement nonlinear stimulus representations and computations. Below, we describe a parsimonious generalization of balanced network theory that allows for nonlinear stimulus representations by allowing excess inhibition without excess excitation.

## Semi-balanced networks implement nonlinear representations in direct correspondence to artificial neural networks of rectified linear units

Note that Eq (3) is only valid if all elements of $\boldsymbol{r}$ it predicts are non-negative. Early work considered a single excitatory and single inhibitory population, in which case positivity of $\boldsymbol{r}$ is assured by simple inequalities satisfied in a large proportion of parameter space [8]. Similarly, in the simulations described above, we constructed $W$ and $W_x$ so that all components of $\boldsymbol{r}$ were positive for all values of $r_{x1}, r_{x2} > 0$.

In networks with a large number of populations, conditions to assure $r \geq 0$ become more complicated and the proportion of parameter space satisfying $r \geq 0$ becomes exponentially small. In addition, we proved that connectivity structures, $W$, obeying Dale's law necessarily admit some positive external inputs, $X > 0$, for which Eq (3) predicts negative rates (see Proof that all connection matrices admit excitatory stimuli that break the classical balanced state in Methods). Hence, the classical notion of excitatory-inhibitory balance cannot be assured by conditions imposed on the recurrent connectivity structure, $W$, alone, but conditions on stimuli, $X$ or $r_x$, are also needed.

While it is possible that cortical circuits somehow restrict themselves to the subsets of parameter space that maintain a positive solution to Eq (3) across all salient stimuli, we consider the alternative hypothesis that Eq (3) and the balanced network theory that underlies it do not capture the full spectrum of cortical circuit dynamics.

To explore spiking network dynamics when Eq (3) predicts negative rates, we considered the same network as above, but changed the feedforward connection probabilities so that Eq (3) predicts positive firing rates only when $r_{x1}$ and $r_{x2}$ are nearly equal. When $r_{x2}$ is much larger than $r_{x1}$, Eq (3) predicts negative firing rates for population $e1$, and vice versa, due to a competitive dynamic.

Simulating the network with $r_{x1} = r_{x2}$ produces positive rates, asynchronous-irregular spiking, and excitatory-inhibitory balance (Fig 1Ei–1Ev, first 500ms). Increasing $r_{x2}$ to where Eq (3) predicts negative rates for population $e1$ causes spiking to cease in $e1$ due to an excess of inhibition (Fig 1Ei–1Ev, last 500ms).

Notably, input currents to populations $e2$ and $i$ remain balanced when $e1$ is silenced (Fig 1Eiv) so the $i$ and $e2$ populations form a balanced sub-network. These simulations demonstrate a network state that is not balanced in the classical sense because one population receives excess inhibition. However,

1. no population receives excess excitation,

2. any population with excess inhibition is silenced, and

3. the remaining populations form a balanced sub-network.

Here, an excess of excitation (inhibition) in population $a$ should be interpreted as $I_a \sim \mathcal{O}(\overline{JK})$ with $I_a > 0$ ($I_a < 0$). The three conditions above can be re-written mathematically in the large $\overline{JK}$ limit as two conditions,

1. $[Wr + X]_a \leq 0$ for all populations, $a$, and

2. If $[Wr + X]_a < 0$ then $r_a = 0$.

These conditions, along with the implicit assumption that $r \geq 0$, define a generalization of the balanced state. We refer to networks satisfying these conditions as "semi-balanced" since they require that strong excitation is canceled by inhibition, but they do not require that inhibition is similarly canceled. Note that the condition $[Wr + X]_a \leq 0$ does not mean that $I_a \leq 0$, but only that $I_a \sim \mathcal{O}(1)$ whenever $I_a \geq 0$ so that $[Wr + X]_a = 0$ in the large $\overline{JK}$ limit, *i.e.*, no excess excitation.

In other words, populations in the semi-balanced state can receive $\mathcal{O}(\overline{JK})$ net-inhibitory input, but if their input is net-excitatory, it must be $\mathcal{O}(1)$. Hence, the semi-balanced state is characterized by excess inhibition, but not excess excitation, to some neural populations. In contrast, the balanced state requires net-input to be $\mathcal{O}(1)$ regardless of whether it is net-excitatory or net-inhibitory, hence no excess excitation or inhibition. Note that firing rates remain $\mathcal{O}(1)$ in both the balanced and semi-balanced states.

How are firing rates related to connectivity and stimulus structure in semi-balanced networks? We proved that firing rates in the semi-balanced state satisfy (see Derivation and analysis firing rates in the semi-balanced state in Methods for a proof)

$$\boldsymbol{r} = [W\boldsymbol{r} + \boldsymbol{X} + \boldsymbol{r}]^+ \tag{4}$$

in the limit of large $\overline{JK}$ where $[x]^+ = \max(0, x)$ is the positive part of $x$, sometimes called a rectified linear or threshold linear function. Eq (4) generalizes Eq (3) to allow for excess inhibition. Note that $\boldsymbol{r}$ satisfies Eq (4) if and only if it satisfies $q\boldsymbol{r} = [W\boldsymbol{r} + \boldsymbol{X} + q\boldsymbol{r}]^+$ for any $q > 0$ (see Derivation and analysis firing rates in the semi-balanced state in Methods for a proof), which explains why terms with different units can be summed together in Eq (4). Even though it is derived in the limit of large $\overline{JK}$, Eq (4) provides an accurate approximation to firing rates in our spiking network simulations (Fig 1Ev, compare dashed to solid).

It is worth noting that the simplest possible semi-balanced network has one inhibitory population and one excitatory population with the excitatory population silenced by the inhibitory population. This would arise when a condition for the positivity of firing rates in a two-population balanced network is violated [8, 11].

Notably, Eq (4) represents a piecewise linear, but globally nonlinear mapping from $\boldsymbol{X}$ to $\boldsymbol{r}$. Hence, unlike balanced networks, semi-balanced networks implement nonlinear stimulus representations (Fig 1Fi). Eq (4) also demonstrates a direct relationship between semi-balanced networks and recurrent artificial neural networks with rectified linear activations used in machine learning [20] and their continuous-time analogues studied by Curto and others under the label "threshold-linear networks" [21–24]. These networks are defined by equations of the form $\tau.\boldsymbol{r} = -\boldsymbol{r} + [U\boldsymbol{r} + \boldsymbol{X}]^+$. Taking $U = W + Id$ where $Id$ is the identity matrix establishes a one-to-one correspondence between solutions to Eq (4) and fixed points of threshold-linear networks or recurrent artificial neural networks. Indeed, we used this correspondence to construct a semi-balanced spiking network that approximates a continuous exclusive-or (XOR) function (Fig 1Fii–1Fiii), which is impossible with linear networks.

Previous work on threshold-linear networks shows that, despite the simplicity of Eq (4), its solution space can be complicated [21–24]: Any solution is partially specified by the subset of populations, $a$, at which $\boldsymbol{r}_a > 0$, called the "support" of the solution. There are $2^n$ potential supports in a network with $n$ populations, there can be multiple supports that admit solutions, and these solutions can depend in complicated ways on the structure of $W$ and $\boldsymbol{X}$. Hence, semi-balanced networks give rise to a rich mapping from inputs, $\boldsymbol{X}$, to responses, $\boldsymbol{r}$.

We proved that, under Eq (2), the semi-balanced state is realized and Eq (4) is satisfied only if firing rates do not grow large as $\overline{JK} \to \infty$ (see Proof that the semi-balanced state is equivalent to bounding rates in Methods for a proof). In other words, Eq (4) and the semi-balanced state it describes are general properties of strongly and/or densely coupled networks (large $\overline{JK}$) with moderate firing rates. To the extent that cortical circuits have large $\overline{JK}$ values and moderate firing rates, therefore, Eq (4) provides an accurate approximation to cortical circuit responses. In summary, our results establish a direct mapping from biologically realistic cortical circuit models to recurrent artificial neural networks used in machine learning and to the rich mathematical theory of threshold-linear networks.

## Semi-balanced network theory is accurate across models and dynamical states

Recently, Ahmadian and Miller argued that cortical circuits may not be as tightly balanced or strongly coupled as assumed by classical balanced network theory [27]. They quantified the tightness of balance by the ratio of total synaptic input to excitatory synaptic input,

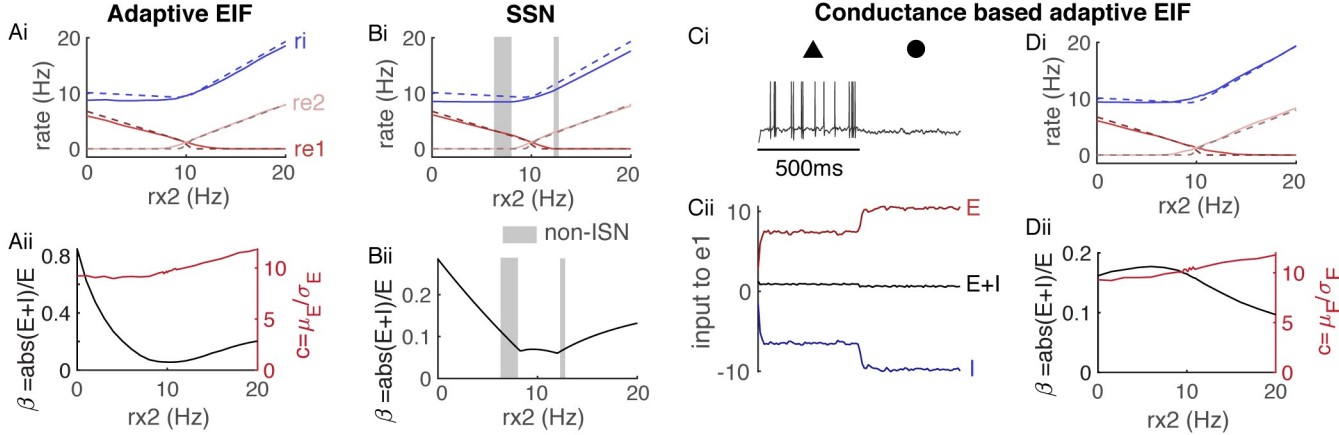

**Fig 2. The semi-balanced approximation is accurate across models and dynamical states. Ai)** Firing rates from simulations (solid) and Eq (4) (dashed) as a function of $r_{x2}$ when $r_{x1} = 10$Hz for the same model as in Fig 1E and 1F. **Aii)** Balance ratio, $\beta$ (black), and coupling strength coefficient, $c$ (red), averaged across all neurons from the simulation in Ai. **Bi-ii)** Same as Ai and Aii, but using dynamical rate equations that implement a supralinear stabilized network (SSN). Gray shaded areas are states in which the network is not inhibitory stabilized. **Ci–ii)** Same as Fig 1E except using a conductance-based model of synapses. **Di-ii)** Same as Ai-ii except using a conductance-based model of synapses. All currents are normalized by the neurons' rheobase.

$\beta = |E + I|/E$ (where $E$ is the mean input current from $e$ and $x$ combined, and $I$ is the mean input from $i$). Small values of $\beta$ imply tight balance, for example $\beta \sim 1/\overline{JK}$ in classical balanced networks. They quantified coupling strength by the ratio of the mean to standard deviation of the excitatory synaptic current $c = \text{mean}(E)/\text{std}(E)$. Strongly coupled networks have large $c$, specifically $c \sim \overline{JK}$. Since Eq (4) was derived in the limit of large $\overline{JK}$, it is only guaranteed to be accurate for sufficiently large $c$, but it is not immediately clear exactly how large $c$ must be for Eq (4) to be accurate.

In our spiking network simulations, Eq (4) was accurate across a range of stimulus values even when $\beta$ and $c$ were in the range deemed to be biologically realistic by Ahmadian and Miller [27] (Fig 2Ai and 2Aii). We conclude that Eq (4) can be a useful approximation for networks with biologically relevant levels of balance and coupling strength.

We next tested the accuracy of Eq (4) against simulations of stabilized supralinear networks (SSNs) proposed and studied by Ahmadian, Miller, and colleagues [27, 29, 30]. In particular, we simulated the three-dimensional dynamical system

$$\tau \dot{r} = -r + k\overline{JK}[Wr + X]_+^2$$

where $[x]_+^2 = ([x]^+)^2$ denotes the square of the positive part of $x$. Simulations of this network with parameters matched to our spiking network simulations show that the network transitioned between an inhibitory-stabilized network (ISN) state to a non-ISN state as $r_{x2}$ varied (Fig 2Bi), which is a defining property of SSNs. Simulations show agreement with Eq (4), even when balance was relatively loose (Fig 2Bi and 2Bii).

A seemingly unrealistic property of semi-balanced networks is that the total mean synaptic current to some populations is $\mathcal{O}(\overline{JK})$ and negative (Fig 1Eiii, black). In our simulations, this strong inhibition clamped the membrane potential to the lower bound we imposed at $-85$mV (Fig 1Eii). The strong inhibitory current is an artifact of using a current-based model of synaptic transmission [31].

In real neurons, the magnitude of inhibitory current is limited by shunting at the inhibitory reversal potential. Repeating our simulations using a conductance-based synapse model to

capture shunting produces similar overall trends to the current-based model (Fig 2Ci and 2Cii) except the mean synaptic input to population $e1$ is no longer so strongly inhibitory (Fig 2Cii, compare to Fig 1Eiii) and membrane potentials of $e1$ neurons still exhibit variability near the inhibitory reversal potential (Fig 2Ci). Eq (4) can be modified to account for conductance-based synapses (see Methods and [15, 32, 33]) and this corrected theory accurately predicted firing rates in our simulations across a range of $c$ and $\beta$ values (Fig 2Di and 2Dii).

## Homeostatic plasticity achieves detailed semi-balance, producing high-dimensional nonlinear representations

So far, we have only considered firing rates and excitatory-inhibitory balance averaged over neural populations. Cortical circuits implement distributed neural representations that are not always captured by homogeneous population averages [34]. Balance realized at single neuron resolution, *i.e.*, where input to each neuron is balanced, is often referred to as "detailed balance" [26, 35]. We therefore use the term "detailed semi-balance" for semi-balance realized at single neuron resolution.

Specifically, generalizing the definitions of population-level balance and semi-balance above, detailed balance is defined by requiring that the net synaptic input to all neurons is $\mathcal{O}(1)$. Detailed semi-balanced only requires neurons' input to be $\mathcal{O}(1)$ when it is net-excitatory. Net-inhibitory input to some neurons will be $\mathcal{O}(\overline{JK})$ in the detailed semi-balanced state. As such, the distribution of total synaptic input to neurons in the semi-balanced state will be left-skewed, indicating strong inhibition to some neurons, but no comparably strong excitation.

To explore detailed balance and semi-balance, we first considered the same spiking network considered above, but with only a single excitatory, inhibitory, and external population (Fig 3A). To model a stimulus with a distributed representation, we first added an extra external input perturbation that is constant in time, but randomly distributed across neurons. Specifically, the time-averaged synaptic input to each neuron was given by the $N \times 1$ vector

$$\vec{I} = \overline{JK}[J\vec{r} + \vec{X}] \tag{5}$$

where $J$ is the $N \times N$ recurrent connectivity matrix and $\vec{r}$ is the $N \times 1$ vector of firing rates. Note that we use the arrow notation, $\vec{I}$, for $N$-dimensional vectors to distinguish them from boldfaced mean-field vectors, like $\mathbf{I}$, that have dimensions equal to the number of populations. We apply the same notational convention to $\vec{r}, \vec{X}$, etc. For a given $\vec{Z}$ the mean $N$-dimensional external input to each neuron is given by

$$\vec{X} = J_x \vec{r}_x + \vec{Z}$$

where, $J_x$ and $\vec{r}_x$ are the feedforward connectivity matrix and external rates. The distributed stimulus, $\vec{Z}$, is defined by

$$\vec{Z} = \sigma_1 \vec{Z}_1 + \sigma_2 \vec{Z}_2$$

where $\vec{Z}_1$ and $\vec{Z}_2$ are standard normally distributed, $N \times 1$ vectors. The vector, $\vec{Z}$, lives on a two-dimensional hyperplane in $N$-dimensional space parameterized by $\sigma_1$ and $\sigma_2$. Hence, $\vec{Z}$ models a two-dimensional stimulus whose representation is distributed randomly across the neural population.

Since we are primarily interested in the encoding of the perturbation, $\vec{Z}$, we could have replaced the spike-based, Poisson synaptic input from the external population with a time-

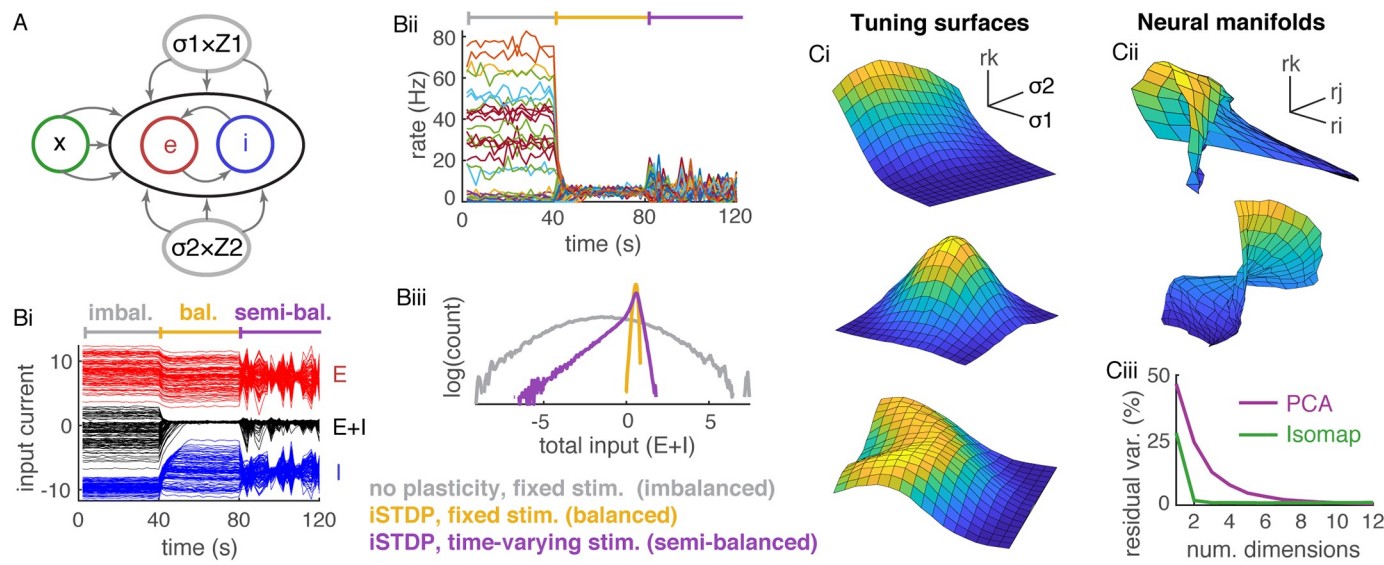

**Fig 3. Detailed imbalance, balance, semi-balance, and distributed neural representations. A)** Network diagram. Same as in Fig 1A except there is just one excitatory and one external population and an additional input $\vec{Z} = \sigma_1\vec{Z}_1 + \sigma_2\vec{Z}_2$. **Bi)** Excitatory (red), inhibitory (blue), and total (black) input currents to 100 randomly selected excitatory neurons averaged over 2s time bins. During the first 40s, synaptic weights and $\sigma_1 = \sigma_2$ were fixed. During the next 40s, homeostatic iSTDP was turned on and $\sigma_1 = \sigma_2$ were fixed. During the last 40s, iSTDP was on and $\sigma_1$ and $\sigma_2$ were selected randomly every 2s. **Bii)** Firing rates of the same 100 neurons averaged over 2s bins. Biii) Histograms of input currents to all excitatory neurons averaged over the first 40s (gray, imbalanced), the next 40s (yellow, balanced), and the last 40s (purple, semi-balanced). **Ci)** Firing rates of three randomly selected excitatory neurons as a function of the two stimuli, $\sigma_1$ and $\sigma_2$ (the neuron's "tuning surface") in a network pre-trained by iSTDP. **Cii)** Three neural manifolds. Specifically, the surface traced out by the firing rates of the three randomly selected neurons as $\sigma_1$ and $\sigma_2$ are varied. **Ciii)** Percent variance unexplained by PCA (purple) and Isomap (green) applied to all excitatory neuron firing rates from the simulation in Ci-ii. Network size was $N = 3 \times 10^4$ in Bi-iii and reduced to $N = 5000$ in Ci-iii to save runtime (see Methods). All currents are normalized by the neurons' rheobase.

constant, DC input to each neuron as in previous work [8]. We chose to keep the spike-based input to add biological realism and to demonstrate the the encoding of $\vec{Z}$ is robust to the Poisson noise induced by the background spike-based input. A more biologically realistic model might encode $\vec{Z}$ in the spike times themselves instead of using an additive perturbation.

Simulations show that this network does not achieve detailed balance or semi-balance: Some neurons receive excess inhibition and some receive excess excitation (Fig 3Bi, first 40s), leading to large firing rates in some neurons (Fig 3Biii) and a broad distribution of total input currents (Fig 3Bii, gray). Indeed, it has been argued previously that randomly connected networks break detailed balance when stimuli and connectivity are not co-tuned [13, 26]. This is consistent with previous results on "imbalanced amplification" in which connectivity matrices with small-magnitude eigenvalues values can break balance when external inputs are not orthogonal to the corresponding eigenvectors [15]. When $J$ is large and random, it will have many eigenvalues near the origin, which can lead to imbalanced amplification if $\vec{X}$ is not orthogonal to the corresponding eigenvectors (see Analysis of detailed imbalance in networks with random structure in Methods for a more precise analysis in terms of singular values).

Previous work shows that detailed balance can be realized by a homeostatic, inhibitory spike-timing dependent plasticity (iSTDP) rule [25, 26]. Indeed, when iSTDP was introduced in our simulations, detailed balance was obtained and firing rates became more homogeneous (Fig 3Bi and 3Bii, second 40s) with a much narrower distribution of total input currents (Fig 3Biii, yellow), indicating detailed balance, at least while $\sigma_1$ and $\sigma_2$ were held fixed.

Of course, real cortical circuits receive time-varying stimuli. To simulate time-varying stimuli, we randomly selected new values of $\sigma_1$ and $\sigma_2$ every 2s (Fig 3Bi and 3Bii last 40s). This

change lead to some neurons receiving excess inhibition in response to some stimuli, but neurons did not receive correspondingly strong excess excitation (Fig 3Bi, black curves last 40s) resulting in a left-skewed distributions of synaptic inputs (Fig 3Biii purple). These results are consistent with a detailed semi-balanced state, which is characterized by excess inhibition to some neurons, but a lack of similarly strong excitation. These results show that detailed semi-balance, but not detailed balance, is naturally achieved in circuits with iSTDP and time-varying stimuli.

To gain a better intuition for why the distribution in (Fig 3Biii, purple) is left-skewed, consider the network with iSTDP and time-varying stimuli. iSTDP changes weights in a way that encourages all excitatory firing rates to be close to a target rate [25] (we used a target rate of 5 Hz). In the presence of a stimulus that varies faster than the iSTDP learning rate, the network cannot achieve the target rates for every neuron at every stimulus. However, the network is pushed strongly away from states with large, net-excitatory input to some neurons because those states produce large firing rates that are very far from the target rates. On the other hand, the network is not pushed as strongly away from states with large net-inhibitory input to some neurons because those states produce firing rates of zero for those neurons, which is not so far from the target rates.

Repeating our simulations in a model with conductance-based synapses shows that shunting inhibition prevents strong inhibitory currents, consistent with evidence that shunting inhibition is prevalent in visual cortex [36], but if currents are measured under voltage clamp then recorded currents are similar to those in Fig 3Bi–3Biii, with excess hyperpolarizing currents in the semi-balanced state (see S1 Fig).

Firing rates in the detailed semi-balanced state are not very broadly distributed (Fig 3Bii, last 40s), which is inconsistent some cortical recordings. Note that the broadness of the firing rate distribution is partly a function of the magnitude of the perturbation strengths, $\sigma_1$ and $\sigma_2$. Also, all of our perturbations lie on a two-dimensional plane, so they could potentially be balanced more effectively by iSTDP than higher dimensional perturbations. Finally, our iSTDP rule used the same target rate for all neurons, which may not be realistic. Stronger perturbations, higher-dimensional perturbations, and variability in target rates, among other factors, could lead to broader firing rate distributions in the detailed semi-balanced state. The width of firing rate distributions for naturalistic stimuli should be considered in future work, but is outside the scope of this study.

We next investigated the properties of the mapping from the two-dimensional stimulus space to the $N$-dimensional firing rate space in the semi-balanced state. We sampled a uniform lattice of $17 \times 17 = 289$ points in the two-dimensional space of $\sigma_1$ and $\sigma_2$ values, simulated a pre-trained network at each stimulus value, and plotted the resulting firing rates of three randomly selected neurons as a function of $\sigma_1$ and $\sigma_2$. The resulting surfaces appear highly nonlinear and multi-modal (Fig 3Ci). Next, we plotted two randomly selected neural manifolds, each defined by the firing rates of three random excitatory neurons. These manifolds also appear highly nonlinear with rich structure (Fig 3Cii). Note that there are over $10^{10}$ such manifolds in the network, suggesting a rich representation of the two-dimensional stimulus. It is worth noting that, due to the presence of plasticity, the same stimulus presented at two different points in time might not have the same firing rate representation.

The nonlinearity of the stimulus representation is more precisely quantified by comparing the results of the dimension reduction techniques isometric feature mapping (Isomap) and principal component analysis (PCA) applied to the sampled firing rates. Both methods attempt to find a low-dimensional manifold in $N$-dimensional rate space near which the sampled rates lie. However, PCA is restricted to linear manifolds (hyperplanes) while Isomap finds nonlinear

manifolds [37]. We applied both methods to the set of all excitatory firing rates across all 289 stimuli from the simulations above.

Despite the fact that firing rates represent 289 points in a 4000-dimensional space, the points lie close to a two-dimensional manifold because they are approximately a function of the two-dimensional stimulus. Applying Isomap shows that the vast majority of the variance was explained by a two-dimensional manifold (Fig 3Ciii, green; 1.76% residual variance at 2 dimensions). However, PCA required more than 8 dimensions to capture the same amount of variance and generally captured less variance per dimension (Fig 3Ciii, purple). This implies that the two-dimensional neural manifold in 4000-dimensional space is nonlinear, *i.e.*, curved, so that it cannot be captured by a two-dimensional plane.

In summary, when networks are presented with time-varying stimuli, iSTDP produces a detailed semi-balance, but not detailed balance. The mapping from stimuli to firing rates is richly nonlinear in the detailed semi-balanced state. We next explore how this nonlinearity improves the computational capacity of the network.

## Nonlinear representations in semi-balanced networks improve classification

To quantify the computational capabilities of spiking networks in the semi-balanced state, we used a network identical to the one from Fig 3 except we replaced the random stimulus, $\vec{Z}$, with a linear projection of pixel values from images in the MNIST data set (Fig 4A, layer 1; see below for description of layer 2). Unlike the 2-dimensional stimuli considered previously, the images live in a 400-dimensional space (20 × 20 pixels).

We first trained inhibitory synaptic weights with iSTDP using 100 MNIST images presented for 1s each. We then presented 2000 images to the trained network and recorded the

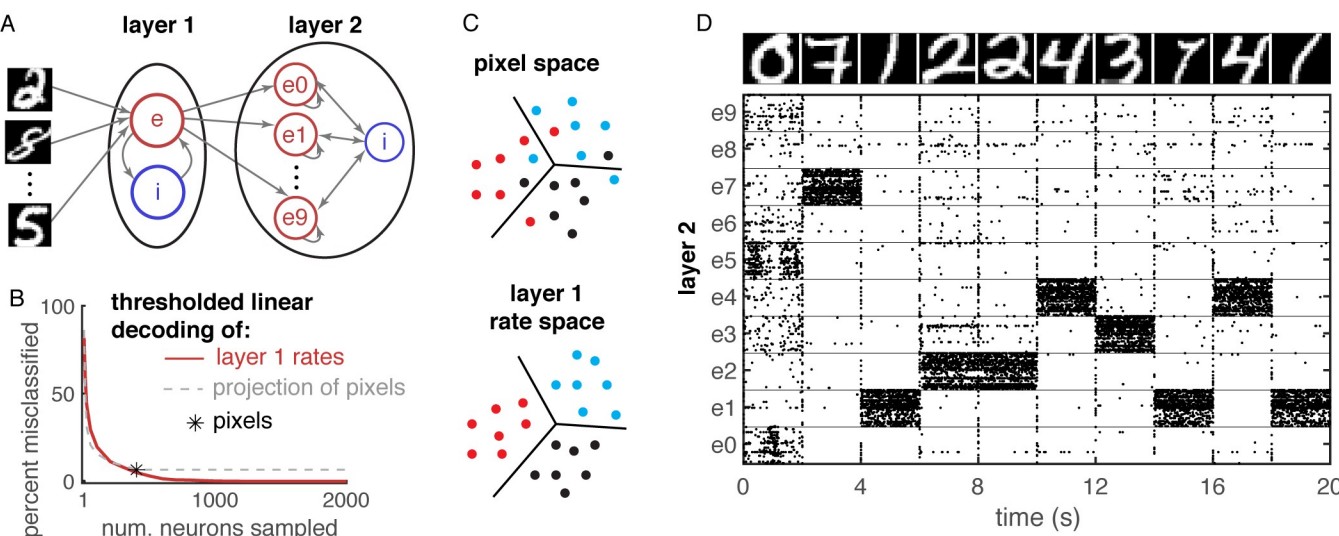

**Fig 4. Nonlinear representations in a semi-balanced network improve classification of MNIST digits. A**) Network diagram. Pixel values provided external input to a semi-balanced network identical to the one in Fig 3, representing layer 1. Layer 2 is a competitive, semi-balance network receiving external input from excitatory neurons in layer 1 with inter-laminar weights trained using a supervised Hebbian-like rule to classify the digits. **B**) Error rate (percent of 2000 images misclassified) of a thresholded linear readout of excitatory firing rates from layer 1 with readout weights optimized to classify the images, plotted as a function of the number, $n$, of neurons sampled (red). Black asterisk shows the error rate of an optimized readout of the $n = 400$ image pixels. Dashed gray shows the error rate of a thresholded linear readout of a random projection of the pixels into $n$ dimensions. The error rate of the rate readout (red curve) is zero for $n \geq 1600$. **C**) Diagram illustrating linear separability in rate space, but not pixel space. Different colors represent different digits. **D**) Raster plot of 500 randomly selected neurons from layer 2 (50 from each population, $ek$) when images at top were provided as external input to layer 1.

firing rates over each stimulus presentation. Applying the same Isomap and PCA analysis used above to these 2000 firing rate vectors confirms that the network implements a nonlinear representation of the images (S2 Fig).

We wondered if the nonlinearity of this representation imparted computational advantages over a linear representation. The 10 different digits (0-9) form ten clusters of points in the 4000-dimensional space of layer 1 excitatory neuron firing rates. Similarly, the raw images represent ten clusters of points in the 400-dimensional pixel space. Can these clusters of points be classified perfectly by thresholding a linear readout?

To answer this question, we defined a linear readout of the 2000 firing rate vectors into 10 dimensions and trained the readout weights to be maximized at the dimension corresponding to the digit's label. Specifically, we defined a $10 \times 4000$ readout matrix, $W_r$ and a 10-dimensional readout vector, $\vec{x} = W_r \vec{r}_e$, where $\vec{r}_e$ is the $4000 \times 1$ vector of excitatory neuron firing rates in layer 1. We then minimized the $\ell^2$ loss function,

$$L = \sum_{i=1}^{2000} \|\vec{x}_i - \hat{x}_i\|_2^2$$

where $\vec{x}_i$ is the readout for MNIST digit $i = 1, \ldots, 2000$ and $\hat{x}_i$ is the one-hot encoding of the label ($\hat{x}_i$ is a $10 \times 1$ vector for which the $j$th element is equal to 1 when $j$ is the $i$th digit's label, and all other elements are zero). We chose a one-hot encoding because it allowed us to test whether digits could be classified by thresholding a linear readout. We chose an $\ell^2$ loss because it can be minimized explicitly without any dependence on hyperparameters.

Using this procedure, we found that all 2000 digits were classified perfectly by thresholding the trained linear readout of firing rates. For comparison, we used the same method to train a linear readout of the 2000 raw MNIST images, treated as vectors in 400-dimensional pixel space. Specifically, we replaced $\vec{r}_e$ above with the pixel-space representation of the images. This analysis revealed that 6.6% of the images were misclassified (Fig 4B, asterisk). Hence, the digits are linearly separable using our procedure in rate space, but not in pixel space (Fig 4C). Hence, the separability in rate space is due to the nonlinearity of the neural representation.

We next investigated how many neurons or encoding dimensions were necessary to achieve linear separability. First, we used the same procedure to train a readout of the $n$ randomly selected layer 1 excitatory neurons and computed the percentage of the 2000 images that were misclassified. The error decreased with $n$ and perfect classification (zero misclassified digits) was achieved for $n \geq 1600$ (Fig 4B, red). Similar results were found by taking a random projection of rates into $n$ dimensions instead of sub-sampling neurons.

To compare the rate space representation to pixel space representation, we projected each raw image randomly into $n$-dimensional space and trained a linear readout. The error of this readout for $n \leq 400$ was similar to the error in rate space (Fig 4B, compare gray dashed to red). However, the error in pixel space saturated to 6.6% at $n = 400$ indicating that a linear projection of pixels into a higher dimensional space does not improve classification (Fig 4B, gray dashed curve saturates at $n = 400$).

These results demonstrate that the nonlinearity of our network can improve the discriminability of stimuli, but they do not address how well the linear readout performs on images that were not used in training. Moreover, the readout weights have mixed sign and do not respect Dale's law. We next considered a downstream spiking network, layer 2, that receives synaptic input from excitatory neurons in layer 1 (Fig 4A). Layer 2 has ten excitatory populations and one inhibitory population. Excitatory populations are coupled to themselves and bi-directionally with the inhibitory population, but do not connect to each other, producing a competitive dynamic.

Our goal was to train feedforward weights from excitatory neurons in layer 1 to those in layer 2 that are strictly positive and encourage the $k$th excitatory population in layer 2 to be most active when layer 1 receives a handwritten digit $k$ as input. We used a simple, Hebbian like learning rule in which the weight from neuron $i$ in layer 1 to neuron $j$ in population $ek$ of layer 2 is increased when neuron $i$ is active during the presentation of digit $k$. This rule is not optimal, but maintains positive weights. We applied the rule to the same 2000 images mentioned above, then tested the performance of the learned weights on 200 images not previously presented to the network. In 72.5% of these 200 test images, the network guessed the correct digit in the sense that population $ek$ in layer 2 had the highest firing rate when digit $k$ was presented (Fig 4D).

## Discussion

We introduced the semi-balanced state, defined by an excess of inhibition without an excess of excitation. This state is realized naturally in networks for which the classical balanced state cannot be achieved and produces nonlinear stimulus representations, which are not possible in classical balanced networks. We established a direct mathematical relationship between semi-balanced networks, artificial neural networks, and the rich mathematical theory of threshold-linear networks. Detailed semi-balance is realized naturally in networks with iSTDP and time-varying stimuli and produces nonlinear stimulus representations that improve the network's computational properties.

An alternative mechanism of nonlinear computations in cortical circuits is given by the theory of SSNs with power-law f-I curves [27, 29, 30] and similar approaches [38]. For large $\overline{JK}$, fixed point firing rates in these models converge to the balanced fixed point, Eq (3), when it is positive. At finite $\overline{JK}$, they implement nonlinearities that are not accounted for by Eq (3). These nonlinearities are necessary to capture some experimentally observed response properties [30]. Indeed, fixed point firing rates in SSNs can be expanded in a series for which Eq (3) is the first term [29]. This expansion is derived under the assumption that rates are positive, which implies that the nonlinearities produced by semi-balance are not present. A more complete theory would combine these two approaches to account for both sources of nonlinearity. Ideally, this approach could produce series expansion for which Eq (4) is the first term instead of Eq (3).

Previous work revealed multi-stability and nonlinear transformations at the level of population averages by balanced networks with short term synaptic plasticity [39]. Future work should consider how the nonlinearities introduced by short term plasticity combine with the nonlinearities introduced by semi-balance.

Classical balanced networks are balanced at the population level, but not necessarily at the level of individual neurons (no detailed balance). While such networks can only perform linear computations at the level of population averages, they can perform nonlinear computations at the level of single neurons and their firing rate fluctuations [40–42]. Cortical circuits do appear to perform nonlinear computations at the population level. For example, population responses to high-contrast visual stimuli add sub-linearly, which can be captured by SSNs [30] and semi-balanced networks (see A semi-balanced network model of contrast dependent nonlinear responses in visual cortex in Methods).

We found that the nonlinearities implemented by semi-balanced networks can improve the separability of MNIST digit representations. Previous work shows that high-dimensional, sparse representations can improve decoding [43]. This could help to understand our empirical results since representations in the semi-balanced state are sparse in the sense that some proportion of neurons are silent for any given stimulus.

We demonstrated that semi-balanced networks can implement a continuous XOR nonlinearity at the population level (Fig 1F) and detailed semi-balanced networks implement more intricate nonlinearities at the resolution of single neurons (Fig 3C and 3D), but we did not consider additional types of nonlinearities. Recordings show that visual cortical neurons exhibit a nonlinearity in which low-contrast visual stimuli sum linearly while high-contrast stimuli sum sub-linearly, a phenomenon that can be reproduced by supralinear stabilized networks (SSNs) [30]. We showed that this type of nonlinearity can also be captured by a simple semi-balanced network that obeys Eq (4) (see A semi-balanced network model of contrast dependent nonlinear responses in visual cortex in Methods). Future work should more completely explore the types of nonlinearities that can be expressed by solutions to Eq (4).

We showed that networks with iSTDP achieve detailed semi-balance and produce nonlinear representations at the level of individual neurons (Figs 3 and 4). However, we do not mean to suggest that iSTDP or balance is responsible for the presence of nonlinear representations or the linear separability of MNIST images in rate space. iSTDP is needed for achieving detailed semi-balance, not nonlinear representations. Indeed, repeating simulations from Figs 3 and 4 without iSTDP gives similar results (see S3 Fig). However, networks without iSTDP are imbalanced at the resolution of individual neurons (detailed imbalance, see Fig 3B, gray). In summary, our results show that networks with iSTDP can produce a form of detailed balance (detailed semi-balance) while still implementing nonlinear representations.

We showed that thresholding a linear readout perfectly classified 2000 MNIST digits encoded in firing rate space, but not pixel space. While optimal linear classification is well-defined for two classes, for example by maximum margin classifiers, there is not one universally optimal way to linearly classify data into several categories. We trained the readouts on a one-hot encoding of the labels using an $\ell^2$ loss. Other types of classifiers could lead to perfect classification in pixel space. For example, support vector machines and artificial neural networks trained with backpropagation perform extremely well on MNIST and could easily obtain perfect classification on a training set of 2000 digits. Also, we found that each pair of digits is separable by a hyperplane in pixel space. Indeed, the binary separability of pairs of digits should be expected by Cover's Function Counting Theorem, which says that perfect binary classification of $m$ random points in $N$ dimensions is possible with high probability when $N$ is large and $m/N < 2$ [44]. Since there are about $m = 200$ images in each class (2000 digits with 10 classes) and the images live in $N = 400$ dimensions, we have and $m/N = 200/400 = 0.5$, implying that the images are well within the margin specified in Cover's Theorem. Our results should not be interpreted to imply that the firing rate representations implemented by our spiking networks are especially well-suited to solving MNIST, but rather that they are just one example of a random, sparse, non-linear representation, which are known to improve discriminability [43]. Indeed, repeating our analysis on a random rectified linear layer (representing an untrained, randomly initialized hidden unit) in place of our spiking network gives similar results (S4 Fig).

One limitation of our approach is that it focused on fixed point rates and did not consider their stability or the dynamics around fixed points. Indeed, fluctuations of firing rates and total synaptic inputs are $\mathcal{O}(1)$ under the scaling of synaptic weights that we used. When a solution to Eq (4) exists, it represents a fixed point of Eq (1) in the $\overline{JK} \to \infty$ limit. The fixed point is stable when all eigenvalues of the Jacobian matrix of Eq (1) evaluated at the fixed point have negative real part. Previous work shows that balanced networks can exhibit spontaneous transitions between attractor states [45] which can be formed by iSTDP [25, 46]. Attractor states in

those studies maintained strictly positive firing rates across populations, keeping the networks in the classical balanced state. This raises the question of whether similar attractors could arise in which some populations are silenced by excess inhibition, putting them in a semi-balanced state. Tools for studying these states, and for studying stability and dynamics more generally, could potentially be developed from the mathematical theory of threshold-linear networks [21–24].

Another limitation is that, in our network trained on MNIST digits, the recurrent connections in the first layer were only trained via an unsupervised iSTDP rule, which is agnostic to the image labels. Hence, the network did not learn a label-dependent representation of the stimuli. Moreover, recurrent excitatory weights were not trained. Future work should consider excitatory synaptic plasticity in the recurrent network and supervised learning rules to learn more informative representations.

The semi-balanced state is defined by an excess of inhibition without a corresponding excess of excitation. This is at first glance consistent with evidence that inhibition dominates cortical responses in awake animals [47]. However, it should be noted that synaptic conductances, not currents, were reported in and they only reported conductances relative to their peaks, not raw conductances [47]. It is therefore difficult to draw a direct relationship of the results in [47] to our results on balance or semi-balance. In addition, we found that the dominance of inhibitory synaptic currents is reduced when shunting inhibition is accounted for (Fig 2Cii and S1 Fig). Hence, due to shunting inhibition, our model does not necessarily predict a strong excess of inhibitory currents in the semi-balanced state. A more precise prediction of our model is that stimuli will silence a subset of neurons through shunting inhibition (Fig 2Ci and 2Cii), consistent with evidence that visual inputs evoke shunting inhibition in cat visual cortex [36]. In addition, if synaptic currents are measured under voltage clamp with the potential clamped sufficiently far between the excitatory and inhibitory reversal potentials, we predict a skewed distribution of currents with a heavier tail of hyperpolarizing versus depolarizing currents (S1 Fig, as in Fig 3Biii purple). These predictions should be tested more directly using *in vivo* recordings.

The relationship between connectivity and firing rates in recurrent spiking networks can be mathematically difficult to derive, which can make it difficult to derive gradient based methods for training recurrent spiking networks (though some studies have succeeded, see for example [48, 49]). The piecewise linearity of firing rates in the semi-balanced state (see Eq (4)) could simplify the training of recurrent spiking networks because the gradient of the firing rate with respect to the weights can be easily computed. This could have implications for the design and training of connectivity in neuromorphic hardware.

In summary, semi-balanced networks are more biologically parsimonious and computationally powerful than widely studied balanced network models. The foundations of semi-balanced network theory presented here open the door to several directions for further research.

## Methods

### Description of models and simulations

We modeled a network of $N$ adaptive EIF neurons with $0.8N$ excitatory neurons and $0.2N$ inhibitory neurons. We chose the adaptive EIF neuron model because it is simple and efficient to simulate while also being biologically realistic [50, 51]. For the current-based model used in all figures except Fig 2B and 2C, the membrane potential of neuron $j = 1, \ldots, N_a$ in population

*a* obeyed

$$\tau_m \frac{dV_j^a}{dt} = -(V_j^a - E_L) + D_T e^{(V_j^a - V_T)/D_T} - w + I_j^a(t)$$

$$\tau_w \frac{dw_j^a}{dt} = -w_j^a$$

with the added condition that each time $V_j^a(t)$ crossed $V_{th} = 0\text{mV}$, a spike was recorded, it was reset to $V_{re} = -72\text{mV}$, and $w_j^a$ was incremented by $B = 0.75\text{mV}$. A hard lower bound was imposed at $V_{lb} = -85\text{mV}$. Other neuron parameters were $\tau_m = 15\text{ms}$, $E_L = -72\text{mV}$, $D_T = 1\text{mV}$, $V_T = -55\text{mV}$, and $\tau_w = 200\text{ms}$. Input was given by

$$I_j^a(t) = \sum_b \sum_k J_{jk}^{ab} \sum_n \alpha_b(t - t_{k,n}^b)$$

where $t_{k,n}^b$ is the $n$th spike of neuron $k$ in population $b$ and $\alpha_b(t) = e^{-t/\tau_b}/\tau_b H(t)$ is an exponential postsynaptic current with $H(t)$ the Heaviside step function. Synaptic time constants, $\tau_b$, were 8/4/10 ms for excitatory/inhibitory/external neurons. Synaptic weights were generated randomly and independently by

$$J_{jk}^{ab} = \begin{cases} j_{ab}/\sqrt{N} & \text{with probability } p_{ab} \\ 0 & \text{otherwise} \end{cases}.$$

In Figs 1C, 1E and 2C, external input rates were $r_x = [15\ 15]^T \text{Hz}$ for the first 500ms and $r_x = [15\ 30]^T \text{Hz}$ for the next 500ms.

In Figs 1 and 2, postsynaptic populations were $a = e1, e2, i$ and presynaptic populations were $b = e1, e2, i, x1, x2$ with $N_{e1} = N_{e2} = 1.2 \times 10^4$, $N_i = 6000$, and $N_{x1} = N_{x2} = 3000$ so that $N = N_{e1} + N_{e2} + N_i = 3 \times 10^4$. Neurons in external populations, $x1$ and $x2$, were not modeled directly, but spike times were generated as independent Poisson processes with firing rates $r_{x1}$ and $r_{x2}$. Connection strength coefficients were $j_{ejek} = 0.375$, $j_{eji} = -2.25$, $j_{iek} = 1.70$, $j_{ii} = -0.375$, $j_{ejxk} = 2.70$, and $j_{ixk} = 2.025\text{mV/Hz}$ for $j, k = 1, 2$. Note that these were scaled by $\sqrt{N}$ to get the actual synaptic weights as defined above. Note that some balanced network studies scale weights by $\sqrt{K}$ instead of $\sqrt{N}$. Since we keep connection probabilities fixed, $K \sim N$, so scaling by $\sqrt{N}$ is equivalent to scaling by $\sqrt{K}$. This choice of synaptic weights produced postsynaptic potential amplitudes between 0.07mV and 0.8mV. Connection probabilities in Fig 1C and 1D were $p_{e1e1} = p_{e2e2} = 0.15$, $p_{e1e2} = p_{e2e1} = 0.05$, $p_{e1x1} = 0.08$, $p_{ix1} = p_{ix2} = 0.12$, and $p_{ab} = 0.1$ for all other connection probabilities. Connection probabilities in Fig 1E and 1F and in Fig 2 were the same except $p_{e1x1} = p_{e2x2} = 0.15$, $p_{e1x2} = p_{e2x1} = 0$, and $p_{ix1} = p_{ix2} = 0.15$.

For Fig 2C and 2D, we used the same model except

$$\tau_m \frac{dV_j^a}{dt} = -(V_j^a - E_L) + D_T e^{(V_j^a - V_T)/D_T} - w - g_{e,j}^a(t)(V - E_e) - g_{e,j}^a(t)(V - E_i)$$

where $E_e = 0\text{mV}$, $E_i = -75\text{mV}$,

$$g_{e,j}^a(t) = \sum_b \sum_k J_{jk}^{ab} \sum_n \alpha_b(t - t_{k,n}^b)$$

with the sum taken over excitatory presynaptic populations ($b = e1, e2, x1, x2$), and

$$g_{i,j}^a(t) = \sum_k J_{jk}^{ai} \sum_n \alpha_i(t - t_{k,n}^i).$$

The excitatory presynaptic weights ($j_{ae1}$, $j_{ae2}$, $j_{ax1}$, and $j_{ax2}$) were the same as above, but divided by ($E_e - V_0$) to account for the change of units. Similarly, presynaptic weights ($j_{ai}$) were divided by ($E_i - V_0$). We took $V_0 = V_T = -55$mV, but the accuracy of the theory did not depend sensitively on this choice. To obtain the dashed curves in Fig 2Di, we used Eq (4), but with the original values of $W$ (those used for the current-based model). This is equivalent to rescaling the conductance-based synaptic weights by ($E_e - V_0$) and ($E_i - V_0$), which is the approximation produced by a mean-field theory derived in previous work [15, 32, 33].

For Fig 2B, we solved $\tau\dot{\boldsymbol{r}} = -\boldsymbol{r} + k\overline{JK}[W\boldsymbol{r} + \boldsymbol{X}]_+^2$ using the forward Euler method with $\boldsymbol{r} = [r_{e1}\ r_{e2}\ r_i]^T$, $\boldsymbol{X} = W_x\,\boldsymbol{r}_x$,

$$
W = \begin{bmatrix} w_{e1e1} & w_{e1e2} & w_{e1i} \\ w_{e2e1} & w_{e2e2} & w_{e2i} \\ w_{ie1} & w_{ie2} & w_{ii} \end{bmatrix},
$$

and

$$
W_x = \begin{bmatrix} w_{e1x1} & w_{e1x2} \\ w_{e2x1} & w_{e2x2} \\ w_{ix1} & w_{ix2} \end{bmatrix}
$$

where $w_{ab} = J_{ab}K_{ab}/\overline{JK} = j_{ab}p_{ab}N_b/\overline{JK}$. We set $k = 10$Hz/(mV)$^2$ which provided a rough match to the sample f-I curves in our spiking network while still exhibiting transitions between ISN and non-ISN regimes. To distinguish between ISN and non-ISN regimes, we computed the Jacobian matrix of the network, checked that all eigenvalues had negative real part (verifying that the fixed point was stable), then checked the eigenvalues of the excitatory sub-matrix of the Jacobian (the matrix with the inhibitory column and row removed). The eigenvalues of the full matrix always had negative real part (the fixed point was always stable). If the eigenvalues of the excitatory sub-matrix had positive real part, we classified the network as an ISN at those parameter values, otherwise it was classified as non-ISN.

For Fig 3, the model was the same as above except there was just one excitatory, one inhibitory, and one external population with $N_e = 0.8N$ and $N_i = N_x = 0.2N$ where $N = 3 \times 10^4$ in Fig 3A and 3B. We reduced network size to $N = 5 \times 10^3$ for Fig 3C because simulations for Fig 3C required 289 simulations for 400s each. The long simulation time, 400s, was needed for accurate estimation of individual neuron's firing rates at each stimulus value, which requires a longer runtime than population averaged rates. The simulation for Fig 3C took around 54 CPU hours and run time grows quadratically with $N$, so a simulation with $N = 3 \times 10^4$ would have taken prohibitively long. Stimulus coefficients in Fig 3B were set to $\sigma_1 = \sigma_2 = 22.5$mV (about 1.4 times the rheobase) for the first 80s and randomly selected from a uniform distribution on $[-30, 30]$mV for the last 40s. In Fig 3C, $\sigma_1$ and $\sigma_2$ values were sampled from a uniform $17 \times 17$ lattice on $[-18, 18] \times [-18, 18]$mV (-18mV to 18mV with a step size of 0.15 mV for each of $\sigma_1$ and $\sigma_2$). Connection probabilities between all populations in Fig 3 were $p_{ab} = 0.1$. Initial synaptic weights were given by $j_{ee} = 37.5$, $j_{ei} = -225$, $j_{ie} = 168.75$, $j_{ii} = -375$, $j_{ex} = 2700$, and $j_{ix} = 2025$mV/Hz as above. Only inhibitory weights onto excitatory neurons ($j_{ei}$) changed, all others were plastic.

The inhibitory plasticity rule was taken directly from previous work [25]. The variables, $x_j^a(t)$, represent filtered spiking activity and are defined by $\tau_x dx_j^a/dt = -x_j^a$ with the added condition that $x_j^a(t)$ was incremented by one each time neuron $j$ in population $a = e, i$ spiked.

After each spike in excitatory neuron $j$, inhibitory synaptic connections onto that neuron were updated by $\Delta J_{jk}^{ei} = -\eta x_k^i(t)$ for all non-zero $J_{jk}^{ei}$. After each spike in inhibitory neuron, $k$, its outgoing synaptic connections were updated by $\Delta J_{jk}^{ei} = -\eta(x_j^e(t) - \alpha)$. We used $\tau_x = 200$ms and $\alpha$ = 2 to get a "target rate" of $r_e^t = \alpha/(2\tau_x) = 5$Hz.

Layer 1 in Fig 4 was identical to the model in Fig 3C (with $N$ = 5000) except the external input was replaced by $\vec{X}_i(t) = \bar{X}_i$ where $\bar{X}_i$ is the mean external input to inhibitory neurons in simulations with an external population (as in previous figures), so the time-varying input to inhibitory neurons was replaced by a time-constant input with the same mean. The external input to excitatory neurons was $\vec{X}_e(t) = \bar{X}_e + \vec{Z}$ where $\vec{Z} = Q\vec{x}$ where $\vec{x}$ is a $400 \times 1$ vector of pixel values in the presented MNIST digit and $Q$ is a $N_e \times 400$ projection matrix where $N_e$ = 4000. We constructed $Q$ so that the $k$th pixel projected to 10 neurons, specifically to neuron indeices $j = 10(k-1) + 1$ through $10k$ with strength $\sigma$. This corresponds to setting $Q_{jk} = \sigma$ for $10(k-1) + 1 \leq j \leq 10k$ and $Q_{jk} = 0$ otherwise. We set $\sigma = 20$mV.

We first trained the inhibitory synaptic weights by presenting 100 MNIST inputs for 1 s each with iSTDP turned on. We then froze the inhibitory weights and presented an additional 2000 MNIST digits for 10 s each and saved the resulting excitatory firing rates for each digit and each excitatory neuron. Weights were frozen for this simulation because the goal is to study the (fixed) representation of digits by the trained recurrent network.

To compute the readout of firing rates from Layer 1, we defined a readout $Y = W_r R_1$ where $\vec{R}_1$ is the $4000 \times 2000$ matrix of the $N_e$ = 4000 Layer 1 excitatory neuron firing rates for each of 2000 MNIST digit inputs, averaged over the 10 s that it was presented to the network. To train the $10 \times 4000$ readout matrix, $W_r$, we minimized the $\ell^2$ (Euclidean) norm between the $10 \times 2000$ matrix, $Y$, and the binary matrix $H$ for which $H(m, n) = 1$ only if digit $n = 1, \ldots,$ 2000 was labeled with $m - 1 = 0, \ldots, 9$. In other words, $H$ is a matrix of one-hot vectors encoding the labeled digit. Since the $\ell^2$ loss is quadratic, the minimizing $W_r$ can be found explicitly. Accuracy was then computed by checking if the maximum index of $Y$ was at the correct digit, i.e., by taking $\tilde{Y}(m, n) = 1$ if $Y(m, n) \geq Y(m', n)$ for all $m = 1, \ldots, 10$. As reported in Results, we obtained perfect accuracy with this procedure, i.e., we obtained $\tilde{Y} = H$ exactly. To compute the readout of pixel values, represented by an asterisk in Fig 4B, we repeated these procedures except we used the $400 \times 1$ vector of pixel values in place of the $4000 \times 1$ vector of excitatory neuron firing rates. For the red curve in Fig 4B, we performed the same procedure, but restricted to a randomly chosen subset of the 4000 excitatory neuron firing rates (subset size indicated on the horizontal axis). For the dashed gray curve in Fig 4B, we used a random projection, $U\vec{x}$, of the pixel values where $\vec{x}$ is the $400 \times 1$ vector of pixel values and $U$ is a $K \times 400$ matrix with $K$ being the number on the horizontal axis of the plot.

Layer 2 in Fig 4 had $N$ = 5000 neurons. The inhibitory population contained $N_i$ = 1000 neurons and there were ten excitatory populations each with 400 neurons. Neurons in the same excitatory population were connected with probability $p_{ejej}$ = 0.1 and neurons in different excitatory populations were connected with probability $p_{ejek}$ = 0 for $j \neq k$. Connection probabilities between the inhibitory population and each excitatory population were $p_{eji} = p_{iej}$ = 0.1. Recurrent connection weights, $j_{ab}$, were the same as for all networks considered above. Layer 2 received feedforward input from Layer 1, i.e., Layer 1 served as the external input population to Layer 2.

Connectivity from Layer 1 to Layer 2 was determined as follows. We first defined a $10 \times 400$ matrix, $U$, with entries $U_{mn} \geq 0$ representing connectivity from neurons in Layer 1 receiving input from pixel $k = n, \ldots, 400$ to neurons in Layer 2 representing digit $m - 1 = 0,$ $\ldots, 9$. We trained these weights on a simulation of Layer 1 with 2000 different MNIST digit inputs. For each digit, if the digit label was $m - 1 = 0, \ldots, 9$, we increased $U_{mn}$ by the sum of all

excitatory firing rates of neurons in Layer 1 receiving input from pixel $m$. In other words, $\Delta U_{mn} = \eta \vec{r}_1 \cdot L$ where $\vec{r}_1$ is a vector of Layer 1 firing rates and $L = [0 \cdots 1 \cdots 0]$ is a $10 \times 1$ vector which is equal to 1 in the place of the labeled digit, *i.e.*, a one-hot vector [20]. We then normalized each column and row of $U$ by its norm. This normalization makes the choice of $\eta$ arbitrary, so we chose $\eta = 1$. The $4000 \times 4000$ feedforward connection matrix, $J^{21}$, from excitatory neurons in Layer 1 to excitatory neurons in Layer 2 was then defined by $J^{21}_{jk} = U_{mn}$ where $m - 1 = 0, \ldots, 9$ is the population to which neuron $j = 1, \ldots, 4000$ belongs and $n = 1, \ldots, 400$ is the pixel from which neuron $k$ receives input. Inhibitory neurons in Layer 2 did not receive feedforward synaptic input, only recurrent input. Since excitatory neurons in Layer 2 are only connected to other excitatory neurons within their population, but all excitatory populations connect reciprocally to the inhibitory population, this creates a winner-take-all dynamic in which the excitatory population with the strongest external input spikes at an elevated rate and suppresses other excitatory populations. Combined with the supervised Hebbian plasticity rule, this creates a dynamic where the network learns to activate population *em* when an image is presented that is similar to training images that were labeled with digit $m$. Fig 4D and the accuracy reported in Results reflects spiking activity in Layer 2 after training of the feedforward weights is turned off.

Code to produce all figures can be found at https://github.com/RobertRosenbaum/SemiBalanceNets/.

## Proof that all connection matrices admit excitatory stimuli that break the classical balanced state

Here, we prove that all connection matrices, $W$, satisfying Dale's law admit some $X$ with positive entries for which some firing rates given by Eq (3) are negative. The theorem relies on the presence at least one excitatory population in the network.

**Theorem 1**. *Suppose W is a real, non-singular $n \times n$ matrix for which each column is either non-negative or non-positive (Dale's law), each column has at least one non-zero element, and there is at least one positive entry in the matrix. Then there exists an $n \times 1$ vector, **X**, with strictly positive entries ($X_j > 0$ for all j) for which the $n \times 1$ vector defined by $r = -W^{-1} X$ has at least one negative entry ($r_j < 0$ for some j).*

*Proof.* Without loss of generality, we can rearrange columns to write $W$ with the non-negative columns first and the non-positive ones next,

$$W = \begin{bmatrix} + & + & \cdots & - & - \\ + & + & \cdots & - & - \\ \cdots & & & & \\ + & + & \cdots & - & - \end{bmatrix}$$

where each + is an element that is $\geq 0$ and each − is $\leq 0$. Now define an $n \times 1$ column vector

$$v = \begin{bmatrix} - \\ - \\ \cdots \\ + \\ + \end{bmatrix}$$

where each − is a negative number, each + is a positive number, there are the same number − entries in $v$ as there are + columns in $W$, and the same number of + entries in $v$ as − entries

in $W$. Finally, define

$$
\begin{aligned}
X &= -W\mathbf{v} \\[4pt]
&= -\begin{bmatrix} + & + & \cdots & - & - \\ + & + & \cdots & - & - \\ \cdots & & & & \\ + & + & \cdots & - & - \end{bmatrix} \begin{bmatrix} - \\ - \\ \cdots \\ + \\ + \end{bmatrix} \\[4pt]
&= \begin{bmatrix} + \\ + \\ \cdots \\ + \\ + \end{bmatrix}
\end{aligned}
$$

In the last expression, each $+$ is a positive number. Note that elements of $X$ cannot be zero because of our assumption that each column of $W$ has at least one non-zero entry.

Now define, $r = -W^{-1} X$ and we must show that $r$ has at least one negative entry. Compute

$$
r = -W^{-1}X = W^{-1}W\mathbf{v} = \mathbf{v}.
$$

Therefore, $r$ has at least one negative entry under our assumption that $W$ has at least one column with non-negative entries.

Note that our proof actually gives infinitely many $X$ that satisfy the theorem, one for each $\mathbf{v}$ having the sign pattern defined in the proof. Moreover, there may exist additional $X$ that are different from the ones generated by our proof.

### Derivation and analysis firing rates in the semi-balanced state

We now prove that Eq (4), which specifies firing rates in the semi-balanced state is equivalent to the two conditions preceding it, which define the semi-balanced state.

**Theorem 2**. *Suppose $W$ is an $n \times n$ matrix and $X$ an $n \times 1$ vector. An $n \times 1$ vector, $r$, satisfies*

A)   $[Wr + X + r]^{+} = r$

*if and only if it satisfies the following three conditions at every index $a = 1, \ldots, n$:*

1. $[Wr + X]_a \leq 0$

2. *If $[Wr + X]_a \leq 0$ then $r_a = 0$*

3. $r_a \geq 0$

*Proof.* We first show that $A$ implies conditions 1–3. Assume $r$ satisfies $A$ and consider some index, $a$. We need to show that 1–3 are all satisfied at $a$. Condition 3 is satisfied because $r_a = [\cdots]^{+} \geq 0$. We still need to prove that conditions 1–2 are satisfied. Note that we either have $r_a = 0$ or $r_a > 0$. First consider the case that $r_a = 0$. Then 2 is satisfied automatically and we only need to prove 1. If $r_a = 0$ then, by A, $[Wr + X]_a^{+} = r_a = 0$ which implies that $[Wr + X] \leq 0$. Now we must consider the case $r_a > 0$. By A, $[Wr + X + r]_a^{+} = r_a > 0$, so the ReLu is

evaluated at its positive part and we can conclude that $r_a = [Wr + X + r]_a = [Wr + X]_a + r_a$. Cancelling the two $r_a$ terms implies that $[Wr + X]_a = 0$. Hence, 1 and 2 are both satisfied. This concludes the proof that A implies 1–3.

Now we must prove that 1–3 implies A. We therefore assume 1–3 and derive A at each index, $a$. By 3, we must have $r_a = 0$ or $r_a > 0$. First assume $r_a = 0$. Then $[Wr + X + r]_a^+ = [Wr + X]_a^+ = 0$ where the last step follows from our assumption of 1. Therefore, $[Wr + X + r]_a^+ = r_a = 0$. Now assume $r_a > 0$. Then, by 1 and 2 combined, we must have $[Wr + X]_a = 0$. Therefore, $[Wr + X + r]_a^+ = [r_a]^+ = r_a$ since $r_a > 0$. This completes our proof.

Note that the condition $r_a \geq 0$ was not explicitly included in the results because it was implicitly assumed. In the first half of our proof, we concluded that $[Wr + X]_a = 0$ wherever $r_a > 0$. This implies that balance is maintained at each population that has a non-zero firing rate, *i.e.*, that the populations with non-zero rates form a balanced sub-network.

The equation $[Wr + X + r]^+ = r$ at first appears awkward because it sums terms with potentially different dimensions: $r$ has dimension 1/time (*e.g.*, units Hz) while $Wr$ and $X$ have dimensions of the neuron model's input current (measured in mV in our model since we normalized by the leak conductance, see Methods). The following theorem clarifies that this combination of dimensions is consistent because one can introduce a scaling factor without changing the solution space.

**Theorem 3**. *Let W be an $n \times n$ matrix and let X and r be $n \times 1$ vectors. The equation*

$$[Wr + X + r]^+ = r \tag{6}$$

*is satisfied if and only if the equation*

$$[Wr + X + cr]^+ = cr \tag{7}$$

*is satisfied for every $c > 0$.*

We first prove that Eq (6) implies Eq (7). Assume Eq (6) is true. Let $a$ be some index. Either $r_a = 0$ or $r_a > 0$. First assume $r_a = 0$. Then $[Wr + X]_a \leq 0$ and $cr_a = 0$. Therefore $[Wr + X + cr]_a^+ = [Wr + X]_a^+ = 0 = cr_a$. Now assume $r_a > 0$. Then $cr_a > 0$ and, as discussed above, we must have $[Wr + X]_a = 0$. Therefore $[Wr + X + cr]_a^+ = [cr_a]^+ = cr_a$. This concludes our proof that Eq (6) implies Eq (7).

We must now prove that Eq (7) implies Eq (6). This is trivial because we can simply take $c = 1$.

## Proof that the semi-balanced state is equivalent to bounding rates

We now prove that for firing rate models, the semi-balanced state is realized if and only if $r \sim \mathcal{O}(1)$ as $\overline{JK} \to \infty$. The proof relies on some reasonable assumptions on the f-I curve, *i.e.*, the function $r = f(I)$.

**Theorem 4**. *Suppose W is a fixed $n \times n$ matrix and X a fixed $n \times 1$ vector. Assume that r and I are $n \times 1$ vectors that depend on $\overline{JK}$ with*

$$I = \overline{JK}[Wr + X]$$

*and*

$$r = f(I)$$

*for all sufficiently large values of $\overline{JK} > 0$. Also assume that f(x) is a non-negative, non-decreasing function for which $\lim_{x \to \infty} f(x) = M$, and $\lim_{x \to -\infty} f(x) = 0$. Here, M can be finite in the case of a*

*saturating or sigmoidal f-I curve, or M = ∞ in the case of an f-I curve that does not saturate. If*

$$\boldsymbol{r}^\infty = \lim_{\overline{JK} \to \infty} \boldsymbol{r}$$

*exists and $r_a^\infty < M$ for all a = 1, . . ., n then*

$$[W\boldsymbol{r}^\infty + \boldsymbol{X} + \boldsymbol{r}^\infty]^+ = \boldsymbol{r}^\infty. \tag{8}$$

*Proof.* Assume $\boldsymbol{r}^\infty = \lim_{\overline{JK} \to \infty} \boldsymbol{r}$ exists and is finite. Then we need to show that it satisfies Eq (8). Specifically, for each index, $a = 1, \ldots, n$, we need to show that

$$[[W\boldsymbol{r}^\infty + \boldsymbol{X}]_a + r_a^\infty]^+ = r_a^\infty$$

where $[W\boldsymbol{r}^\infty + \boldsymbol{X}]_a$ is the *a*th index of $W\boldsymbol{r}^\infty + \boldsymbol{X}$. Let $a \in \{1, \ldots, n\}$ be an arbitrary index and define

$$c = \lim_{\overline{JK} \to \infty} \boldsymbol{I}_a \overline{JK}.$$

Note that

$$c = \lim_{\overline{JK} \to \infty} [W\boldsymbol{r} + \boldsymbol{X}]_a = [W\boldsymbol{r}^\infty + \boldsymbol{X}]_a$$

exists and is finite by assumption.

We first argue that $c \leq 0$. To show this, we will assume that $c > 0$ and prove a contradiction. If $c > 0$ then

$$\lim_{\overline{JK} \to \infty} \boldsymbol{I}_a = \lim_{\overline{JK} \to \infty} \overline{JK} c = \infty$$

and therefore

$$r_a^\infty = \lim_{\overline{JK} \to \infty} f(\boldsymbol{I}_a) = M$$

which contradicts our assumption that $r_a^\infty < M$ for all *M*. We may conclude that $c \leq 0$. We now break the proof into two cases: $c = 0$ and $c < 0$.

*Case 1*: $c = 0$.

We have $c = [W\boldsymbol{r}^\infty + \boldsymbol{X}]_a = 0$, so

$$[[W\boldsymbol{r}^\infty + \boldsymbol{X}]_a + r_a^\infty]^+ = [r_a^\infty]^+$$

but $r_a^\infty \geq 0$ at all indices, *a*, because $\boldsymbol{r} = f(\boldsymbol{I}) \geq 0$ at all $\overline{JK}$ and $\boldsymbol{r}^\infty = \lim_{\overline{JK} \to \infty} \boldsymbol{r}$. Therefore,

$$[[W\boldsymbol{r}^\infty + \boldsymbol{X}]_a + r_a^\infty]^+ = [r_a^\infty]^+ = r_a^\infty.$$

This completes Case 1.

*Case 2*: $c < 0$.

We have $c = [W\boldsymbol{r}^\infty + \boldsymbol{X}]_a < 0$, so

$$\lim_{\overline{JK} \to \infty} \boldsymbol{I}_a = \lim_{\overline{JK} \to \infty} \overline{JK} c = -\infty.$$

Therefore,

$$r_a^\infty = \lim_{\overline{JK} \to \infty} f(\boldsymbol{I}_a) = \lim_{\boldsymbol{I}_a \to -\infty} f(\boldsymbol{I}_a) = 0.$$

As a result,

$$[[W\boldsymbol{r}^\infty + \boldsymbol{X}]_a + \boldsymbol{r}_a^\infty]^+ = [[W\boldsymbol{r}^\infty + \boldsymbol{X}]_a]^+ = 0 = \boldsymbol{r}_a^\infty.$$

because $[W\boldsymbol{r}^\infty + \boldsymbol{X}]_a = c < 0$ and $\boldsymbol{r}_a^\infty = 0$. This completes Case 2.

## Analysis of detailed imbalance in networks with random structure

We now show that the balanced state is generally broken in large networks with random structure. First consider the equation

$$\vec{I} = J\vec{r} + \vec{X}$$

where $\vec{I}$ is the $N \times 1$ vector of synaptic inputs to neurons in a network of size $N$, $\vec{X}$ is the external source of input, $\vec{r}$ are the neurons' firing rates, and $J$ is the $N \times N$ connectivity matrix. In classical balanced networks, $J_{jk} \sim \mathcal{O}(1/\sqrt{N})$ and $\vec{X}_j \sim \mathcal{O}(\sqrt{N})$. Balance at single neuron resolution is achieved when $\vec{r}_j \sim \mathcal{O}(1)$ and $\vec{I}_j \sim \mathcal{O}(1)$ for all $j$. By the equation for $\vec{I}$, this requires cancellation between $\vec{X}_j \sim \mathcal{O}(\sqrt{N})$ and

$$[J\vec{r}]_j = \sum_{k=1}^{N} J_{jk} r_k$$

at each index, $j$, and therefore requires that $[J\vec{r}]_j \sim \mathcal{O}(\sqrt{N})$ for all $j$. We argue here that, under natural conditions on the properties of static connectivity matrices, $J$, and high-dimensional inputs, $\vec{X}$, balance at single neuron resolution is impossible. In other words, it is impossible to have both $\vec{r}_j \sim \mathcal{O}(1)$ and $I_j \sim \mathcal{O}(1)$ for all $j$.

To get an intuition for why this is true, note that if $J$ is a large matrix with some randomness and some order in its structure, then $J$ will tend to have a small number of large, $\mathcal{O}(\sqrt{N})$, singular values, but the bulk of the singular values will be randomly distributed and $\mathcal{O}(1)$. This is related to the fact that large matrices with random structure have most of their eigenvalues within a circle of fixed radius around the origin of the complex plane. The singular vectors corresponding to these $\mathcal{O}(1)$ singular values represent directions, $\vec{v}$, in which $\|J\vec{v}\| \ll \|\vec{v}\|$. Therefore, $J\vec{r}$ is small (specifically $\mathcal{O}(1)$) when projected onto the subspace spanned by these singular vectors. On the other hand, if these singular vectors point in random directions with respect to $\vec{X}$ then the projection of $\vec{X}$ onto this subspace is much larger (specifically $\mathcal{O}(\sqrt{N})$). Therefore, $J\vec{r}$ cannot cancel $\vec{X}$ within this subspace, so the projection of $\vec{I} = J\vec{r} + \vec{X}$ onto this subspace is large (specifically, $\mathcal{O}(\sqrt{N})$), which implies a break in balance.

A more rigorous development of this conclusion follows. We begin with the general theorem, then discuss why the assumptions in the theorem are naturally satisfied by randomly connected balanced network models with static synapses and why the conclusions of the theorem imply a lack of balance at single-neuron resolution.

**Theorem 5**. *For each sufficiently large positive integer $N$, suppose that $\vec{I}$, $\vec{r}$, and $\vec{X}$ are $N \times 1$ vectors and $J$ is an $N \times N$ matrix satisfying*

$$\vec{I} = J\vec{r} + \vec{X}$$

*and*

$$\|X\| \sim \mathcal{O}(N)$$

*where $\|\cdot\|$ denotes the Euclidean 2-norm. Let $J = U\Sigma V^T$ be the singular value decomposition of $J$*

*with singular values listed in ascending order ($\sigma_{j+1} \geq \sigma_j$ where $\sigma_j = \Sigma_{jj}$ is the jth singular value, note that this is backward from the standard convention). Suppose that there exists a constant, $\rho_0 > 0$, that does not depend on N and a positive integer $N_0 \leq N$ with $N_0 \sim \mathcal{O}(N)$ and*

1. $\sigma_j \leq \rho_0$ *for j = 1, . . ., $N_0$ and*

2. $\displaystyle\sum_{j=1}^{N_0} \frac{(\vec{U}_j \cdot \vec{X})^2}{\|\vec{U}_j\|^2 \|\vec{X}\|^2} = \mathcal{O}(1)$

*where $U_j$ is the jth column of U. Then it is impossible to have both $\|\vec{r}\| \leq \mathcal{O}(\sqrt{N})$ and $\|\vec{I}\| \leq \mathcal{O}(\sqrt{N})$.*

*Proof.* Assume that $\|\vec{r}\| \leq \mathcal{O}(\sqrt{N})$ and $\|\vec{I}\| \leq \mathcal{O}(\sqrt{N})$. We must derive a contradiction. Multiplying $\vec{I} = J\vec{r} + \vec{X}$ on both sides by $U^T$ gives

$$U^T\vec{I} = \Sigma V^T\vec{r} + U^T\vec{X}.$$

Now let $U_0$ and $V_0$ be the $N \times N_0$ matrices composed of the first $N_0$ columns of U and V respectively and let $\Sigma_0$ be the $N_0 \times N_0$ diagonal matrix formed by the first $N_0$ rows and columns of $\Sigma$. Then

$$\|U_0^T\vec{I}\| = \|\Sigma_0 V_0^T\vec{r} + U_0^T\vec{X}\|. \tag{9}$$

Now note that

$$\|U_0^T\vec{I}\| \leq \|\vec{I}\| \leq \mathcal{O}(\sqrt{N})$$

and, similarly,

$$\|\Sigma_0 V_0^T\vec{r}\| \leq \rho_0\|V_0^T\vec{r}\| \leq \rho_0\|r\| \leq \mathcal{O}(\sqrt{N}).$$

Now compute

$$\|U_0^T\vec{X}\|^2 = \sum_{j=1}^{N_0}(U_j \cdot \vec{X})^2 = \mathcal{O}(\|X\|^2) = \mathcal{O}(N^2)$$

by assumption 2 above and the fact that $\|U_j\| = 1$. Therefore,

$$\|U_0^T\vec{X}\| = \mathcal{O}(N).$$

This contradicts Eq (9) since the left hand side is no greater than $\mathcal{O}(\sqrt{N})$ and the right hand side is $\mathcal{O}(N)$.

The following lemma and discussion explains why the conclusion of Theorem 5—that it is impossible to have both $\|\vec{r}\| \leq \mathcal{O}(\sqrt{N})$ and $\|\vec{I}\| \leq \mathcal{O}(\sqrt{N})$—implies a break of balance.

**Lemma 1**. *Suppose $\vec{u}$ is an $N \times 1$ vector for each positive integer N. If $|\vec{u}_j| \leq \mathcal{O}(1)$ as $N \to \infty$ for all j then $\|\vec{u}\| \leq \mathcal{O}(\sqrt{N})$.*

*Proof.* We have that

$$\|\vec{u}\| = \sqrt{\sum_{j=1}^{N} u_j^2} \leq \sqrt{\sum_{j=1}^{N} \mathcal{O}(1)} = \mathcal{O}(\sqrt{N}).$$

Therefore, the conclusion of Theorem 5 implies that it is impossible to have $r_j \sim \mathcal{O}(1)$ and $I_j \sim \mathcal{O}(1)$. In other words, the conclusion of the theorem implies that $|r_j| \to \infty$ for some j or

$|I_j| \to \infty$ for some $j$ (or both). Note that if we assume that $|r_j| \to \infty$ implies $|I_j| \to \infty$ (as would be the case if $r_j = f(I_j)$ for some finite function, $f$) then the conclusion of the theorem implies that $|I_j| \to \infty$, *i.e.*, there's a break in balance.

We now explain why the assumptions of Theorem 5 are reasonable for balanced network models. First note that, by the same reasoning used to prove Lemma 1, the assumption that $\|\vec{X}\| \sim \mathcal{O}(N)$ is implied by assuming that $X_j \sim \mathcal{O}(\sqrt{N})$, which is a defining assumption of balanced networks. If $\vec{X}$ is a random vector, for example, then $\|\vec{X}\| \sim \mathcal{O}(N)$ if the mean and standard deviation of the elements, $X_j$, are $\mathcal{O}(\sqrt{N})$.

The assumption that there are $N_0 \sim \mathcal{O}(N)$ singular values with $\sigma_j \leq \rho_0$ is a common property of random matrices. The eigenvalues of random matrices are more widely studied than the singular values, but note that the singular values are the square roots of the eigenvalues of the symmetric non-negative definite matrix, $J^T J$. Most balanced network models assume a blockwise Erdös-Renyi structure on $J$ with one block for each pair of $n$ populations (so $n^2$ blocks in all). More specifically, most balanced network models have $n = 2$ population, one excitatory and one inhibitory. The eigenvalues and singular values of these block-wise Erdös-Renyi matrices have a $n$ values that are $\mathcal{O}(\sqrt{N})$, corresponding to the mean-field directions of the block-wise structure. The remaining values are randomly distributed in a region of radius $\mathcal{O}(1)$ (a circle in the complex plane for eigenvalues, an interval for singular values, which are real). Hence, if there are $n \sim \mathcal{O}(1)$ blocks, then there are $N_0 = N - n \sim \mathcal{O}(N)$ singular values with $\mathcal{O}(1)$ magnitude. The corresponding singular vectors, $U_j$, are random, unit vectors, *i.e.*, they are direction vectors with random directions.

The final assumption of Theorem 5 is that

$$\sum_{j=1}^{N_0} \frac{(\vec{U}_j \cdot X)^2}{\|\vec{U}_j\|^2 \|X\|^2} = \mathcal{O}(1).$$

Since $\|U_j\| = 1$ and $\sum_{j=1}^{N_0} (\vec{U}_j \cdot X)^2 = \|U_0^T X\|^2$, this is equivalent to

$$\|U_0^T X\|^2 = \mathcal{O}(\|X\|^2)$$

Note that, since the columns of $U$ form an orthonormal basis,

$$\|X\|^2 = \|U_0^T X\|^2 + \|U_1^T X\|^2$$

where $U_1$ is the $N \times (N - N_0)$ sub-matrix of $U$ formed by the largest $N - N_0$ columns of $U$ (those omitted from $U_0$). Therefore, assumption 2 in the theorem is equivalent to assuming that $\lim_{N \to \infty} \|U_1^T X\|^2 / (\|X\|^2) \neq 1$, *i.e.*, that there is some variability in $\vec{X}$ that is not asymptotically parallel to the structured part of $U$. So, unless $\vec{X}$ is nearly perfectly parallel to the low-dimensional, structured part of $J$, assumption 2 Theorem 5 would be satisfied.

## A semi-balanced network model of contrast dependent nonlinear responses in visual cortex

In this Appendix, we demonstrate that a simple semi-balanced network can implement a nonlinearity observed in visual cortical circuits in which low-contrast stimuli add linearly and high-contrast stimuli add sub-linearly [30]. We consider a simple model of two visual receptive fields, each associated with an excitatory and an inhibitory population. This gives four populations altogether: $e_1$, $e_2$, $i_1$, and $i_2$ where $e_1$ is the excitatory population at receptive field 1, etc.

We posit a mean-field connectivity matrix of the form

$$W = \begin{bmatrix} 10 & 5 & -60 & -30 \\ 5 & 10 & -30 & -60 \\ 50 & 25 & -100 & 50 \\ 25 & 50 & -50 & -100 \end{bmatrix} mV \cdot ms$$

This matrix represents connectivity that is two times as strong between populations in the same receptive field compared to populations in opposite receptive fields.

A low-contrast stimulus to the first receptive field is modeled by external input of the form

$$X_1^{low} = [0.4 \ 0.3 \ 0.15 \ 0.05]^T mV$$

so that populations $e_1$ and $i_1$ receive stronger external input than populations $e_2$ and $i_2$. Similarly, a low-contrast stimulus to receptive field 2 is modeled by

$$X_2^{low} = [0.3 \ 0.4 \ 0.05 \ 0.15]^T mV.$$

A low-contrast stimulus to both receptive fields is modeled by summing these two stimuli to obtain

$$X_{1+2}^{low} = X_1^{low} + X_2^{low} = [0.7 \ 0.7 \ 0.2 \ 0.2]^T mV.$$

Firing rates predicted by semi-balanced network theory can be computed by solving Eq (4) to obtain

$$\begin{aligned} r_1^{low} &= [11.67 \ 7.67 \ 7.5 \ 3.5]^T Hz \\ r_2^{low} &= [7.67 \ 11.67 \ 3.5 \ 7.5]^T Hz \\ r_{1+2}^{low} &= [19.33 \ 19.33 \ 11 \ 11]^T Hz \end{aligned}$$

for stimuli $X_1^{low}$, $X_2^{low}$, and $X_{1+2}^{low}$ respectively. It is easy to check that $r_{1+2}^{low} = r_1^{low} + r_2^{low}$, so the stimuli add linearly at low contrast.

High contrast stimuli are modeled by

$$\begin{aligned} X_1^{high} &= [0.8 \ 0.3 \ 0.25 \ 0.1]^T mV \\ X_2^{high} &= [0.3 \ 0.8 \ 0.1 \ 0.25]^T mV \\ X_{1+2}^{high} &= X_1^{high} + X_2^{high} = [1.1 \ 1.1 \ 0.35 \ 0.35]^T mV \end{aligned}$$

which give rates

$$\begin{aligned} r_1^{high} &= [32.5 \ 0 \ 18.75 \ 0]^T Hz \\ r_2^{high} &= [0 \ 32.5 \ 0 \ 18.75]^T Hz \\ r_{1+2}^{high} &= [29.66 \ 29.66 \ 17.166 \ 17.166]^T Hz \end{aligned}$$

respectively. Since $r_{1+2}^{high} < r_1^{high} + r_2^{high}$ even though $X_{1+2}^{high} = X_1^{high} + X_2^{high}$, these high-contrast stimuli add sub-linearly.

## Supporting information

**S1 Fig. Balance and semi-balance at single-neuron resolution in a model with conductance-based synapses. A–C)** Same as Fig 3Bi–3Biii except that a conductance-based model was used for synapses. Synaptic currents from population $a$ were measured by $-g_a(t)(V(t) - E_a)$. **D–E)** Same as A–B except "effective" synaptic currents were measured by $I_a(t) = -g_a(t)(V_0 - E_a)$ where we chose $V_0 = -55\text{mV}$, but results did not depend sensitively on the choice of $V_0$. This defines a notion of effective balance and semi-balance in terms of a balance or semi-balance between the effective currents, instead of actual currents. Effective semi-balance and dominance of effective inhibition is an experimentally testable prediction of our model. (PDF)

**S2 Fig. Dimensionality of layer 1 firing rates in the model from Fig 4.** Same as Fig 3Ciii except IsoMap and PCA were applied to firing rates of layer 1 neurons from the model in Fig 4. (PDF)

**S3 Fig. Nonlinear representations without iSTDP. A)** Same as Fig 3Ciii except the network was not trained by iSTDP. **B)** Same as Fig 4B except the network was not trained by iSTDP. (PDF)

**S4 Fig. Classification of MNIST digit representations with a random rectified linear layer.** Same as Fig 4B except the dotted blue curve was added which represents the same as the red curve except the firing rate representation was replaced by a representation in which the raw pixels were projected randomly into 4000 dimensions, then passed through a rectified linear function. Specifically, the pixels were multiplied by a $400 \times n$ matrix of standard normal numbers (the same matrix for each digit) then passed through the function $f(x) = [x]^+ = \max(x, 0)$. (PDF)

## Author Contributions

**Conceptualization:** Cody Baker, Vicky Zhu, Robert Rosenbaum.

**Data curation:** Cody Baker, Vicky Zhu.

**Formal analysis:** Cody Baker, Vicky Zhu, Robert Rosenbaum.

**Funding acquisition:** Robert Rosenbaum.

**Investigation:** Cody Baker, Vicky Zhu, Robert Rosenbaum.

**Software:** Cody Baker, Robert Rosenbaum.

**Supervision:** Robert Rosenbaum.

**Visualization:** Cody Baker, Vicky Zhu, Robert Rosenbaum.

**Writing – original draft:** Cody Baker, Vicky Zhu, Robert Rosenbaum.

**Writing – review & editing:** Cody Baker, Vicky Zhu, Robert Rosenbaum.

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
