## [Decision Letter · Decision Letter 0]

17 Jun 2020

Dear Dr. Rosenbaum,

Thank you very much for submitting your manuscript "Nonlinear stimulus representations in neural circuits with approximate excitatory-inhibitory balance" for consideration at PLOS Computational Biology.

As with all papers reviewed by the journal, your manuscript was reviewed by members of the editorial board and by several independent reviewers. In light of the reviews (below this email), we would like to invite the resubmission of a significantly-revised version that takes into account the reviewers' comments.

We cannot make any decision about publication until we have seen the revised manuscript and your response to the reviewers' comments. Your revised manuscript is also likely to be sent to reviewers for further evaluation.

Sincerely,

Bard Ermentrout

Associate Editor

PLOS Computational Biology

Samuel Gershman

Deputy Editor

PLOS Computational Biology

Reviewer's Responses to Questions

**Comments to the Authors:**

Reviewer #1: This is a nice mathematical analysis of an extension to standard "balanced network" theory that permits imbalance in the form of strong inhibition, but not excitation. This modification permits a richer set of solutions, in particular solutions that are nonlinear functions of the input. My comments are primarily on presentation and improving the part on learning and classification.

Comments:

1) The improvement on MNIST is nice, even though MNIST is a simple benchmark by modern standards. However, the results could be more convincing. Part of the reason is that we don't have a comparison to standard methods, and it's hard to do so because the authors are using a different setup (fewer images, lower-dimensional pixel space). A few suggestions:

a) Could the authors also report the performance of their approach applied to a random projection of the data into a space of the same dimensionality followed by a ReLU (a standard random one-hidden-layer network)? Even better would be to also show a network trained with backprop, which presumably would outperform both approaches.

b) The authors should also describe their "optimal linear readout" more carefully. It sounds like they're minimizing an l2 loss between a linear projection of the firing rates and the one-hot output. But this is not what is typically considered an "optimal" readout, namely a maximum-margin classifier (i.e. support vector machine). The authors should see if their results change if they use an SVM classifier.

2) The mathematical notation would benefit from a re-read and could be cleaned up. As a few comments:

Line 95: r_x should be bolded

Line 98: average(.) and mean(.) are both used in the paper, use consistent standard terminology

Both bold and arrow notation is used for vectors, stick with bold.

I' m not a fan of the "e1" and "e2" and "x1" and "x2" notation. Why not just 1 and 2 for the excitatory populations and x and y for the external populations? This one is not so crucial, more of an aesthetic point.

Minor comments:

-In Figure 2, I wasn't able to understand the expression for \\beta = |E+I|/E at first because the brackets for the absolute value look the same as a capital I.

-Theorem 2 of the S2 appendix seems to start with some missing characters.

-Line 743: "directions in which J only points weakly" -- this is a bit confusing, since J is a matrix, not a vector.

-Line 466: The statement that artificial neural networks often use sigmoids instead of ReLUs is a bit out of date -- ReLUs are standard now.

Reviewer #2: ## Overview

Networks with E/I dynamic balance are a central theme in theoretical neuroscience. Originally, they were developed to explain the observed variability in the activity of single neurons using network effects. The main idea is that individual synapses are random and relatively strong (1/sqrt(N)); this leads to temporal and spatial heterogeneity in synaptic currents. If the network respect Dale’s law, it leads to a situation where the total inhibitory and excitatory synaptic current are strong, but cancel each other to allow O(1) net current —i.e., balance. A direct result of this model is that population activities and inputs must obey a linear relation. This linearity has long puzzled researchers since it seemingly constrains the network to perform linear computations **using population averages**. In this work, the authors show that the Balance Conditions do not need to be met strictly. Instead, some populations can be allowed to have excess inhibitory input current, effectively silencing them. The result is that the linear relation becomes a switch-linear system: linear relation between component, but the participating component change in a nonlinear manner, opening the door to nonlinear computations. In particular, neural populations become effective Rectified Linear Units (ReLU), which have broad applications in machine learning. The authors show that the analogy is robust to parameter choice, underlying neuron dynamics (current or conductance-based), and architecture. The theory is backed by simulation and provides an example for using the framework in a basic machine learning setting.

## Strengths and novelty

The authors offer a possible solution for an open question: Can dynamically balanced networks be useful in nonlinear neural computation? Here are my main takeaways from this paper:

* The authors observe that when the number of populations increases, the volume in parameter space obeying the Balance Equations becomes exponentially small. This finding is in contrast to van Vreeswijk and Sompolinsky’s original work, which showed that Balanced Networks are stable for a wide range of parameters when considering two parameters).

* By allowing a population to be very polarized, their activity is silenced, effectively turning the linear Balance Equations to nonlinear switch-linear equations

* The study shows that a semi-balanced network can serve as a nonlinear kernel enabling the separation of linearly non-separable inputs using a linear readout. It is a concrete example of nonlinear computation that, to my knowledge, has never been shown before for Balanced Networks.

* The authors show that a local synaptic learning rule (iSTDP) can lead to semi-balanced networks.

* The results are robust and can be generalized to more realistic neurons. Furthermore, I think the performance is robust to noise, though the authors do not state that explicitly (more on that below).

* The theory provides testable predictions. For example, the distribution of synaptic currents should show a heavy tail toward hyperpolarizing.

## Criticism and questions

* The premise of this study is that Balance Networks entails a linear relation of the mean firing rates of different populations (the Balance Equations), thus cannot perform nonlinear operations. While the population averages keep a linear relation, the high dimensional activity is not necessarily linear. It is reasonable that computations in a balanced state are carried through higher statistics (e.g., local fluctuations) while keeping the mean activity steady.

* The authors ignore the fluctuations altogether in the paper. With their normalization (line 489), the fluctuations are expected to be O(1). It is not apparent to me if there are further restrictions on the connectivity when multiple populations are concerned, and what will be the stability criteria. While the authors acknowledge not studying stability (line 433), its absence is a significant drawback of this report.

* The nonlinear computations presented are equivalent to a nonlinear expansion of the input space into higher dimensions. This procedure has been shown to improve the network classification performance on otherwise linearly-inseparable data [Babadi and Sompolinsky, Neuron, 2014]. It is not clear to me why and to what extent the balance is required for that. Are the authors assuming balance as a given state, or are they suggesting that balance is beneficial? I think a simple network of nonlinear units will do just that without balance. Is the balance a way to get an effective firing-rate based unit out of spiking neurons? Is this a way to reduce readout fluctuations [Deneve and Machens, Nat. Neuro. 2016] and provide robustness to external noise? I think the authors should address the role and importance of (semi)-balance in their results.

### Minor issues and comments

* In the model of the 2D input (Figure 3, and paragraph starting line 265). What is the role of the external population x? is it to simulate background noise? Following van Vreeswijk and Sompolinsky, I believe the model should work without the noisy input, with only a DC input. The strength of the input from the noisy external population should provide evidence to the robustness to noise.

* In the paragraph starting at line 370, the authors ask what the number of neurons needed to achieve separability is. It seems to be more of an observation on the data instead of on the model. As a comparison, they could use the theoretical results by Cover on the dimensions needed to allow linear classification (P=2N, where N is the number of neurons, and P is the number of points).

* The 2-layers model (Fig 4) seems analogous to sparse expansion with feedforward inhibition (Babadi and Sompolinsky, 2014]. The authors should relate their models to the known results.

* line 459: “Recurrent spiking neural networks are notoriously difficult to train...”. Training spiking networks has been done both for Deep Networks [e.g., Zenke and Ganguli, 2017] and in recurrent networks [Nicola and Clopath, Nature 2017]

* Line 466: “Artificial recurrent neural networks for machine learning often use sigmoidal activation functions instead of the rectified linear activations typically used in feedforward networks because the unboundedness of rectified linear units make recurrent networks susceptible to instabilities and large activations”. RNN with ReLU units can be stabilized even if it is unbound, and in particular, with balance [Kadmon and Sompolinsky, PRX, 2015]. Stabilization is achieved by adequately normalizing the weight variance near the transition to chaos. The same also applies to Deep Networks with ReLU [Poole et al. 2016, arXiv:1606.05340] and with tanh activation [Pennington et al. 2017, arXiv:1711.04735]

## Conclusions and recommendations

I find the bottom line of this work to be interesting and fitting for publication in PLoS Comp. Bio. In particular, this study provides an example of how a balanced (or semi-balanced) network can perform nonlinear computations. The idea that the linear relations of the Balance Equations can present rectified linear nonlinearity and, thus, promote nonlinear computations is smart and useful.

Nevertheless, the authors should address the issues presented above before publication. I think the results are correct, and none of my comments should falsify the study. Yet clarification on how this study related to previously known results and possible limitations will help clarify the message here.

Reviewer #3: In this manuscript, the authors address the issue of the mechanisms of

generation of non-linearities in strongly coupled networks of

excitatory and inhibitory neurons. In the popular balanced

network model, average excitatory and inhibitory firing rates depend

linearly on external inputs in the large coupling limit, which does

not seem to fit with a broad range of experimental observations. A

number of studies have proposed in recent years different scenarios to

account for non-linearities in such networks: (1) When coupling

strength is sufficiently far from the strong coupling limit, then

strong non-linearities appear that have been argued to match

qualitatively data (refs [28,29,30,38]) (2) Non-linearities could be

induced by synaptic mechanisms (ref [39]). In this manuscript, the

authors proposed a third scenario, that takes advantage of the

rectification inherent in neuronal activity. In this `semi-balanced'

scenario, different sub-populations of the network could be suppressed

(and so out of balance), leading to the possibility of non-linear

computations. They characterize analytically this scenario, showing

among other things that for any non-singular connectivity matrix,

there exists external stimuli that push the network in the

semi-balanced state, and that steady state firing rates in such

networks are equivalent to steady-state firing rates in

threshold-linear networks. They also argue that iSTDP leads to

additional interesting properties, and that such networks can be used

as an intermediate layer to transform non-linearly separable inputs

into linearly separable ones. Overall, this is a stimulating and

timely paper that I believe should ultimately be published in

PLOSCB. However, I have a number of suggestions to improve the paper,

and concerns that I believe the authors should address before the

paper is accepted for publication.

Major comments

1. The critique of the classical balanced state found in the

introduction seems a bit overblown.

- l.49-50: Satisfying constraints on parameters in the standard balanced

network is very easy (only a few inequalities on parameters), and

there is no known model in which satisfying these constraints has been

found to be difficult. Furthermore, homeostatic mechanisms such as the

iSTDP mechanism proposed in this manuscript is an easy way out in case

initial parameters do not satisfy the inequalities. All in all, this

issue does not seem at all to be a `critical shortcoming'.

- l.61: It is not obvious why the results in ref.[19] would be

incompatible with the standard balanced state picture - in that paper

the authors mention that inhibitory conductances are much larger than

excitatory conductances in the awake state, but this is not

necessarily incompatible with a balance of currents.

2. The authors remain quite vague in the paper about what types of

non-linearities are observed in cortex, and whether their model can

reproduce these specific types of non-linearities (as opposed to a

generic ability of exhibiting non-linear responses). For instance, in

visual cortex one typically sees linear summation of two stimuli at

low contrast, but sublinear summation at high contrast (see

ref.~30). Can the model exhibit these non-linearities?

3. In the section on semi-balance in networks with heterogeneous

inputs and iSTDP is somewhat confusing. In particular, while the role

of iSTDP in producing detailed (semi) balance is clear, its role in

producing non-linear representations is not.

The classic balanced network model features linearity of the AVERAGE

firing rates as a function of input, but single neuron responses can

be quite non-linear (van Vreeswjik and Sompolinsky, in Methods and

Models in Neurophysics, 2005, Elsevier). Also, strongly heterogeneous

inputs will themselves push neurons outside of their linear range in

the absence of iSTDP. The question therefore arises whether similar

non-linear representations could arise in networks with no iSTDP. To

check this, the authors could repeat the analysis described in panels

C in networks with no iSTDP.

Another issue with iSTDP is that distributions of firing rates

recorded in cortex are typically very wide, and much closer to the no

iSTDP case than the case with iSTDP. It has been shown that such broad

distributions are an automatic by-product of the random connectivity

in balanced networks (e.g.~Roxin et al 2011). From this point of view,

the standard balanced network model seems more realistic that the one

with iSTDP where the distribution of rates is narrow (unless one puts

by hand a broad distribution of target firing rates).

Finally, the authors should provide some intuition why the distribution of

total inputs is left skewed. Is this because of the specific

functional form of the equation in line 544, which produces large

jumps in the case of excess excitation but not in the case of excess

inhibition ($x^E_j$ can grow to really large values but cannot go

below zero)?

4. The authors show that, when inputs evolve dynamically,

the learned representation is nonlinear. The authors should comment on

the fact that, depending on the history of the stimuli presented over

time, the same stimulus can have completely different

representations. This seems to have major implications for sensory

encoding.

5. In the section: "Nonlinear representations in semi-balanced

networks improve computations", they show that linear separability of

images improves when they are fed into a network of E-I neurons. As in

the previous section, it is unclear what role iSTDP plays exactly. As

in point 3 above, I believe they should compare results obtained with

and without iSTDP.

Minor comments:

- It could be a good idea to mention that the simplest possible

semi-balanced network has just two populations, one silent E

population and an active I population.

- In line 109 (page 3), ref[8]: Technically, the van

Vreeswijk-Sompolinsky papers did not use spiking networks. To my

knowledge, a linear relationship between firing rates and external

inputs in networks of spiking neurons was first derived by Brunel in

2000.

- In line 110 (page 3). N not defined.

- Figure 2B, How was the ISN property tested?

- In the methods, it would be good to provide some justification of

why this specific single neuron model was chosen, and of the choice of

parameters:

l.498, synaptic weights of order $1/\\sqrt{N}$: Shouldn't they be of

order $1/sqrt{K}$? Of course, the choice would be important only if

the authors analyzed how network behavior changes as a function of $K$

or $N$, something which is not done in this paper.

l.508-515: The description of how synaptic strengths are computed in the

conductance-based version is confusing - I believe the current-based

weights should be divided by the driving force to get the

conductance-based weights, not the opposite. This confusion is present

both in the first and the last sentence of the paragraph.

The description of the connectivity from pixels to layer 1 is also

confusing and does not seem to match the ratio of number of neurons to

pixels. With 400 pixels, the indicated procedure would work if there

were 40,000 neurons, but there are only 4,000 excitatory

neurons. Shoudn't 100 be replaced by 10? Also,the value of

$\\sigma$ is not specified.

**Have all data underlying the figures and results presented in the manuscript been provided?**

Reviewer #1: Yes

Reviewer #2: Yes

Reviewer #3: Yes

PLOS authors have the option to publish the peer review history of their article (what does this mean?). If published, this will include your full peer review and any attached files.

Reviewer #1: No

Reviewer #2: Yes: Jonathan Kadmon

Reviewer #3: No
---

## [Decision Letter · Decision Letter 1]

24 Jul 2020

Dear Dr. Rosenbaum,

We are pleased to inform you that your manuscript 'Nonlinear stimulus representations in neural circuits with approximate excitatory-inhibitory balance' has been provisionally accepted for publication in PLOS Computational Biology.

Best regards,

Bard Ermentrout

Associate Editor

PLOS Computational Biology

Samuel Gershman

Deputy Editor

PLOS Computational Biology

Reviewer's Responses to Questions

**Comments to the Authors:**

Reviewer #1: The authors have addressed my major concerns. As a final note: the authors are correct that using support-vector machines for multi-class classification is not straightforward (and usually ensembles of binary SVMs are used). But there is still a much better choice than an L2 loss for multi-class classification, namely minimizing the cross-entropy loss. However, this is sufficiently minor that I don't think the authors need to do it unless they want to.

Reviewer #2: The authors have addresses my concerned, and I recommend the paper be published in PLoS Comp. Bio. in its current form.

Reviewer #3: I am happy with the revision.

**Have all data underlying the figures and results presented in the manuscript been provided?**

Reviewer #1: Yes

Reviewer #2: Yes

Reviewer #3: Yes

PLOS authors have the option to publish the peer review history of their article (what does this mean?). If published, this will include your full peer review and any attached files.

Reviewer #1: No

Reviewer #2: **Yes: **Jonathan Kadmon

Reviewer #3: No

---

## [Editor Report · Acceptance letter]

14 Sep 2020

PCOMPBIOL-D-20-00650R1

Nonlinear stimulus representations in neural circuits with approximate excitatory-inhibitory balance

Dear Dr Rosenbaum,

I am pleased to inform you that your manuscript has been formally accepted for publication in PLOS Computational Biology. Your manuscript is now with our production department and you will be notified of the publication date in due course.

With kind regards,

Matt Lyles
